



# Profiling pollen and biomass burning particles over Payerne, Switzerland using laser-induced fluorescence lidar and *in situ* techniques during the 2023 PERICLES campaign

Marilena Gidarakou[1], Alexandros Papayannis[1,2], Kunfeng Gao[3], Panagiotis Gidarakos[4], Benoît Crouzy[5], Romanos Foskinis[1,2,5], Sophie Erb[5], Cuiqi Zhang[3], Gian Lieberherr[5], Martine Collaud Coen[5], Branko Sikoparija[6], Zamin A. Kanji[3], Bernard Clot[5], Bertrand Calpini[5], Eugenia Giagka[1], Athanasios Nenes[2,7]

[1]Laser Remote Sensing Unit (LRSU), Physics Department, National Technical University of Athens, Zografou, Greece
[2]Laboratory of Atmospheric Processes and Their Impacts, School of Architecture, Civil and Environmental Engineering, École Polytechnique Fédérale de Lausanne, Lausanne, Switzerland
[3]Institute for Atmospheric and Climate Science, ETH Zürich, Zurich, Switzerland
[4]Athena Research Center, Greece
[5]Federal Office of Meteorology and Climatology MeteoSwiss, Payerne, Switzerland
[6]BioSense Institute – Research Institute for Information Technologies in Biosystems, University of Novi Sad, Novi Sad, Serbia
[7]Centre for Studies of Air Quality and Climate Change, Institute of Chemical Engineering Sciences, Foundation for Research and Technology Hellas, Patras, Greece

Correspondence to: Alexandros Papayannis (alexandros.papagiannis@epfl.ch), Athanasios Nenes (athanasios.nenes@epfl.ch)

**Abstract.** Vertical profiles of pollen and biomass burning particles were obtained at a semi-rural site at the MeteoSwiss station near Payerne (Switzerland) using a novel multi-channel elastic-fluorescence lidar combined with *in situ* measurements during the spring 2023 wildfires and pollination season during the PERICLES (PayernE lidaR and Insitu detection of fluorescent bioaerosol and dust partiCLES and their cloud impacts) campaign. Pollen particles were detected near ground (up to 2 km height), showing strong fluorescence backscatter coefficients $b_F$ at 355 nm ($b_F$ ~2 x $10^{-4}$ $Mm^{-1}sr^{-1}$ to 8 x $10^{-4}$ $Mm^{-1}sr^{-1}$). Smoke plumes from Canada and Germany were detected at higher altitudes (3-5 km) and showed lower $b_F$ values compared to those from pollen particles near ground. In situ measurements and *in vivo* fluorescence spectra were used to classify pollen particles near ground. Ice nucleating particle (INP) concentrations relevant for mixed-phase clouds showed high concentrations at warm temperatures, characteristic of the contribution of biological particles to the INP population. This was further supported by the correlation of INPs at –14°C with WIBSABC particles, indicating a contribution from fluorescent biological aerosol particles, while INPs at –20°C were more strongly linked to coarse-mode dust. The analysis of $b_F$ values across two European LIF lidar stations revealed that aged air masses containing smoke particles can show a ~50% reduction of these values during their transport in the free troposphere (3-5 km) possibly due to photochemical aging and mixing with other non-fluorescent particles.



## 1. Introduction

Aerosol particles are key components of the atmosphere as they play a crucial role of the Earth radiation budget and climate through scattering and absorbing incoming solar radiation, modulating cloud formation and development, precipitation and the hydrological cycle (Bellouin et al., 2020; Dubovik et al., 2006; Intergovernmental Panel On Climate Change (IPCC), 2023;

Lohmann, 2017; Nenes and Seinfeld, 2003; Stevens and Feingold, 2009). However, large uncertainties remain on the role of aerosols on Earth's climate and radiation forcing (Bjordal et al., 2020; Chen et al., 2022; IPCC, 2023; Lohmann and Lesins, 2002; Seinfeld et al., 2016; Watson-Parris and Smith, 2022). Biological aerosols (or "bioaerosols") comprise an important and largely understudied type of atmospheric aerosols and include pollen grains, fungal spores, viruses, bacteria, algae, and cell fragments. It is now established that bioaerosols impact ecosystem, cloud formation and possibly climate (Fröhlich-Nowoisky

et al., 2016; IPCC, 2023; Violaki et al., 2021, 2025; Wilson et al., 2015); however, there is still a large gap in the scientific understanding of the interaction and co-evolution of life and climate in the Earth system (Fröhlich-Nowoisky et al., 2016). Bioaerosols can act as cloud condensation nuclei (CCN) (Mikhailov et al., 2021; Petters et al., 2009) and ice nucleating particles (INPs) in mixed-phase clouds (Gao et al., 2024, 2025). The latter can occur at very warm temperatures, and thus bioaerosols can initiate ice multiplication that leads to rapid glaciation, storm intensification and extreme precipitation (Gao et

al., 2025; Lohmann et al., 2016; O'Sullivan et al., 2015).

Pollen grains are ubiquitous in the atmosphere during the pollination period of different flowering plants like trees, grasses and other herbaceous plants. Climate induced-warming over the last decade during winter and spring lead to an earlier onset and increased intensity with a resulting extension in the pollen season (Beggs, 2016; Glick et al., 2021; Lake et al., 2017; Ziska et al., 2019) and exposure of humans to pollen allergens (Albertine et al., 2014; Martikainen et al., 2023; Zemmer et al., 2024).

Pollen fluorescence spectra analysis is increasingly used to monitor these changes and provide valuable information on biogenic aerosol classifications (pollen, bacteria, fungi) and other aerosol types containing biological materials (e.g. BB and dust aerosols) using different light-induced fluorescence techniques based on UV-light sources (LEDs, flash lamps, lasers; (Huffman et al., 2020; Sauvageat et al., 2020; Tummon et al., 2024) and aloft (Reichardt et al., 2025; Richardson et al., 2019; Veselovskii et al., 2023, 2024).

Vertical profiles of fluorescent agents in the lower stratosphere were first detected by Immler et al. (2005) using laser remote sensing (lidar). Later, Sugimoto et al. (2012) detected the presence of fluorescent dust particles with a multi-channel lidar spectrometer (420-510 nm) using laser radiation at 355 nm as excitation source. Few years later, Saito et al. (2018, 2022) detected pure pollen particles very close to the source, based on the analysis of lidar LIF signals and reference pollen fluorescence *in vitro* spectra obtained with 355 nm excitation. Richardson et al. (2019) were the first to classify different types

of mixed bioaerosols (pollen, fungi, bacteria) in the lower troposphere by deconvoluting lidar LIF bioaerosol fluorescence spectra using a 32-channel fluorescence spectrometer (based on *in vitro* reference fluorescence spectra for specific pollen, fungi, and bacteria) excited at 266 and 355 nm. Veselovskii et al. (2020) combined Mie-Raman and fluorescence (444-487 nm) lidar techniques to detect the presence of fluorescent smoke and dust particles in the troposphere, while later they



characterized atmospheric layers of smoke, dust, pollen and urban particles through a combination of linear particle
depolarization ratio, fluorescence coefficient and capacity vertical profiles using single-channel (detection at 444-470 nm)
(Veselovskii et al., 2021, 2022a, b) and 6-channel (438, 472, 513, 560 and 614 nm) fluorescence lidar systems based
(Veselovskii et al., 2023, 2024, 2025).

The present study aims to further extend the LIF lidar technique from near ground to the free troposphere by combining the
observation systems of Richardson et al. (2019) and Veselovskii et al. (2021). This approach will enable us to classify pollen
types near ground (below 300 m above ground level-a.g.l. by using 30° slant laser emission along a nearby mountain upslope)
and to characterize atmospheric layers of pollen and smoke in the free troposphere during the PERICLES (PayernE lidaR and
In situ detection of fluorescent BB, bioaerosol and dust partiCLES and their cloud impacts) campaign at the Payerne,
Switzlerand MeteoSwiss station (46.822° N, 6.941° E, 491 m a.s.l) from May to December 2023. The main objective of this
campaign was to understand the spatio-temporal variability of different types of bioaerosols (pollen, BB, dust) in the Planetary
Boundary Layer (PBL) and Free Troposphere aloft (typically up to 2-5 km a.s.l.) and their potential role in cloud formation
through a combination of fluorescence LIF lidar, radiosonde profiling, and in-situ (ground-level) characterization of
bioaerosol, air pollutants, chemical composition and INP measurements. Other objectives of this campaign aimed to map the
diversity and sources of bioaerosols in the lower atmosphere to provide insights on the mechanisms controlling their
concentration near ground, to validate existing pollen forecast models (ICON model) (Zängl et al., 2015) on a semi-rural site.
Although the full PERICLES dataset contains a large amount of aerosol episodes, we focus on two, characteristic of long- and
near-range transport of BB and pollen particles (occurred during May and June 2023 using a multi-instrumental approach.

A description of the experimental site and the *in situ* and remote sensing instrumentation implemented during the PERICLES
campaign is given in Section 2, while, in Section 3 we present the methodology used to calibrate the lidar fluorescence channel
for the retrieval of the fluorescence backscatter coefficient, and the fluorescence spectra clustering/analysis for pollen
classification near ground. In Section 4 we present the observational results regarding the vertical profiles of the aerosol elastic
backscatter and bioaerosols (smoke and pollen) fluorescence backscatter and capacity, obtained during two case studies. In
Section 5 we discuss our results in conjunction with similar data obtained in France and Germany during the same period and
summarize the main results obtained during this campaign.

## 2. Experimental site and Instrumentation

### 2.1 Site location

The PERICLES campaign took place at the Payerne MeteoSwiss station, close to Payerne, Switzerland (46.82° N, 6.94° E,
altitude 490 m a.s.l.), between 1 May and 20 December 2023. The site is located on a hilly terrain within the Swiss lowlands
at a semi-urban aera are surrounded by grassland and several farmlands, and close to a forest ecosystem between the Jura
mountains and the Alps (Fig. 1a-c). The position of the *in situ* instrumentation and the lidar is described in Fig. 1(d-e).The
station is member of the Aerosol, Clouds, and trace gases research Infrastructure-ACTRIS (www.actris.eu) and hosts a suite





of air quality measuring instrumentation of the National Air Pollution Monitoring Network-NABEL of Switzerland, which in turn is part of the European Monitoring and Evaluation Programme (EMEP), and stores its data at EUROAIRNET, which was established by the European Environment Agency (EEA).

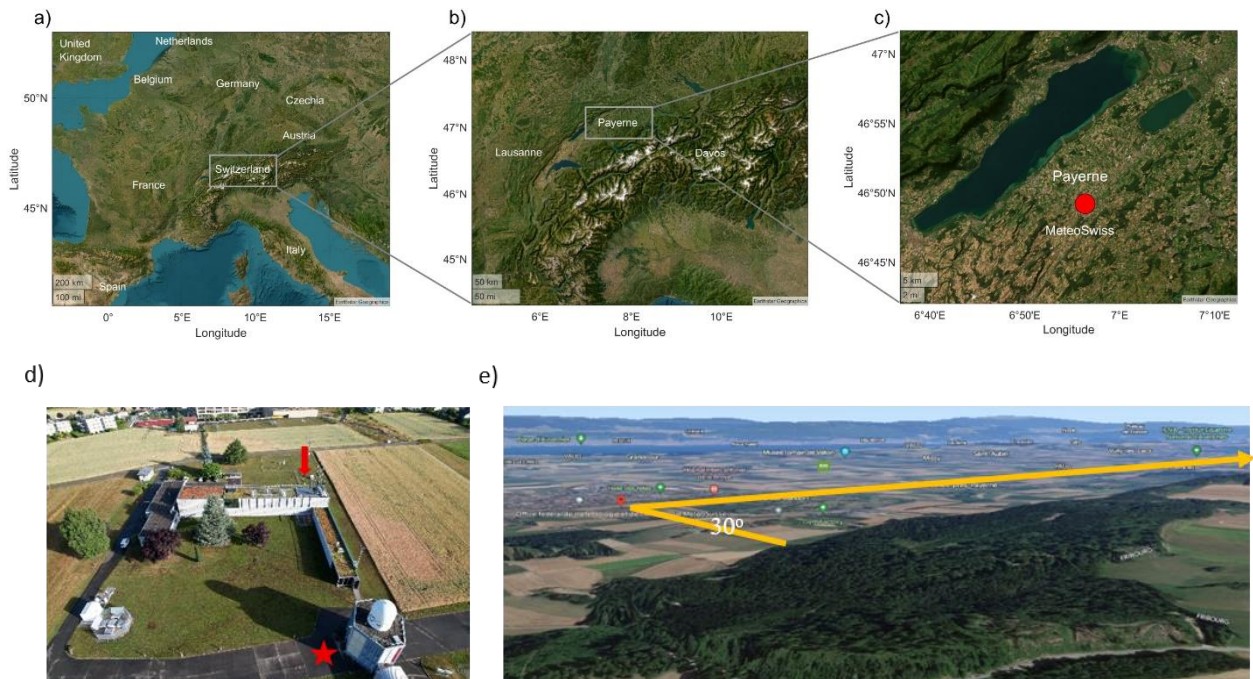


**Figure 1. (a)** The study area of Switzerland **(b)** the sub-domain over Payerne **(c)** the experimental site of MeteoSwiss in Payerne **(d)** MeteoSwiss in Payerne the roof of MeteoSwiss is indicated by the vertical arrow and the position of the lidar by an asterisk **(e)** the emitted laser beam (30⁰ elevation angle to the ground) passing over the nearby hill by 50-200 m (right). © Google Maps 2025.

### 2.2 Instrumentation

To comprehensively understand the meteorology and general atmospheric dynamical conditions, aerosols, characteristics and potential impact on cloud formation, a large suite of *in situ* and remote sensing instruments was deployed at the Payerne site (Table S1). In this paper we focus on the detection of *in situ* pollen and bioaerosols, as well as the vertical distribution of aerosol concentration, type (bioaerosol, BB) and chemical composition to understand the sources of particles that influence the aerosol in the air column. *In-situ* bioaerosol measurements were based on automatic light- (or/laser)-induced fluorescent

and holographic instrumentation, in parallel with Hirst-type pollen traps that provide a golden standard for comparison (Crouzy et al., 2016). Furthermore, investigations on the role of bioaerosols as INPs provided important insights on their contribution in cloud formation (Gao et al., 2024, 2025) . The remote sensing instrumentation for the detection of non- and fluorescent aerosols was based on a single-wavelength elastic and laser-induced fluorescence lidar system developed by NTUA, EPFL and FORTH, with single or multiple detection channels. All *in situ* measuring instruments were deployed at the rooftop of the

MeteoSwiss building (8 m above ground level), while the remote sensing instruments were located ~70 m away and with an



unobstructed view of the *in situ* measuring devices. Meteorological data (wind speed and direction, relative humidity, temperature) affecting low tropospheric air mass dynamics and mixing were obtained using operational MeteoSwiss weather stations, wind lidar, radar and local radiosoundings.

### 2.2.1 In situ instrumentation

During the PERICLES campaign, a wide variety of in situ instrumentation was deployed, such a Swisensat, a Hirst-type volumetric trap, a Wideband Integrated Bioaerosol Sensor (WIBS-5-NEO), an Aethalometer AE33, a Time of-Flight Aerosol Chemical Speciation Monitor (ToF-ACSM), a high-flow rate impinger (Coriolis®micro-μ), as well as an offline droplet freezing assay called DRoplet Ice Nuclei Counter Zurich (DRINCZ). The SwisensPoleno (Swisens AG-Lucerne, Switzerland) is an all-optical automatic pollen monitoring device using scattered light measurements, digital holography, and induced light-

induced fluorescence to identify and count different pollen types. Firstly, particle size and velocity are determined using scattered light from two trigger lasers. Following this first assessment, two 200×200-pixel images are obtained using digital holography at 90° perspectives with a resolution of 0.53 μm. Following the holography module, two light-emitting diode (LED) light sources at 280 and 365 nm, and a laser source at 405 nm excite the sampled particles and produce fluorescence. These spectra are recorded across five different measurement windows (335–380, 415–455, 465–500, 540–580, and 660–690

nm). A blower maintains a sampling airflow rate of 40 L min$^{-1}$, and a virtual impactor unit that concentrates aerosol particles with diameters larger than 5 μm (Sauvageat et al., 2020). Inference allowing the identification of individual particles is performed in the device itself (embedded computer). A three-step classifier is used: firstly, the size and the overall shape allow to filter out most non-biological coarse aerosol, secondly, convolutional neural networks are used for taxa identification and as a final step. The classification is evaluated following the procedure of Crouzy et al. (2022). Hirst-type volumetric trap (Hirst,

1952) was also utilized to identify and count pollen in complement to the automatic monitors at roof level. The Hirst sampler is calibrated at a flow rate of 10L min$^{-1}$ and captures pollen grains on a rotating drum covered with Melinex film coated in silicon fluid (Galán et al., 2014; O'Connor et al., 2014; Savage et al., 2017). Each 48 mm on the drum represents 24 hours of sampling resulting in a full rotation of the drum in a week. Once the drum is fully rotated the pollen grains are counted, offline, under a light microscope. Daily pollen counts are obtained by running four longitudinal sweeps along the 24-hour slide and

identifying each pollen type. This manual counting procedure complements the automatic monitors used and serves as a reference measurement, as it allows for the identification of 47 different pollen taxa, albeit at the cost of a reduced temporal resolution and sampling.

The WIBS-5–NEO (Droplet Measurement Technologies, LLC., USA), sampling aerosol particles from an omnidirectional total inlet, was employed in parallel with SwisensPoleno, Rapid-E, and Hirst. The WIBS was used to detect and classify

fluorescent biological aerosol particles (FBAPs) by discriminating the fluorescent properties and optical sizes of particles. WIBS first counts and measures particle sizes between 0.5 and 30 μm using a 635 nm laser beam. WIBS triggers the fluorescence emission of particles at 280 and 370 nm wavelengths and then records their fluorescent signals in two wavebands (310–400 nm and 420–650 nm, respectively). Thus, fluorescent signals at three different channels can be recorded, including



FL1 channel detecting the fluorescence emissions in the waveband 310–400 nm of particles excited at 280 nm, FL2 channel
and FL3 channel detecting the fluorescence emissions in the waveband 420–650 nm of particles excited at 280 and 370 nm
respectively (O'Connor et al., 2014; Savage et al., 2017). The three channels can detect different biologic fluorophores:
tryptophan-containing proteins, NAD(P)H co-enzymes and riboflavin (Kaye et al., 2005; Savage et al., 2017), which are
ubiquitous in microbes (Pöhlker et al., 2012). A particle only carrying one type of fluorophores and showing fluorescence in
only one of the three channels will be classified to a type of WIBS$_A$ (FL1), WIBS$_B$ (FL2) or WIBS$_C$ (FL3). Similarly, a
fluorescent particle detected by two channels will be recorded as an WIBS$_{AB}$ (FL1 and FL2), WIBS$_{AC}$ (FL1 and FL3) or
WIBS$_{BC}$ (FL2 and FL3) particle. WIBS$_{ABC}$ refers to particles having three types of fluorophores and is reported to have the
highest probability of being biological origins among all types of particles (Gao et al., 2024, 2025; Hernandez et al., 2016;
Savage et al., 2017). In addition, the size distribution of aerosol particles recorded by WIBS was used to calculate the number
concentration of total aerosol particles (0.5-30 µm), termed WIBS$_{total}$, and coarse mode particles (2.5-30 µm), termed
WIBS$_{coarse}$.

The mass concentration of elemental black carbon (eBC) and organic carbon (OC), sampled through a PM$_{10}$ cut-off inlet, was
monitored by the Aethalometer AE33 (Magee Scientific, US) working at 880 nm with a time resolution of 1 min (Eleftheriadis
et al., 2009; Stathopoulos et al., 2021) and then averaged on an hourly basis. The chemical composition of non-refractory
aerosols, including organics, sulphate (SO$_4^{2+}$), nitrate (NO$^{3-}$), ammonium (NH$_4^+$) and chloride (Cl$^-$), was measured by the ToF-
ACSM (Aerodyne Research Inc., USA) with a time resolution of 10 min (Fröhlich et al., 2015; Zografou et al., 2024). The
number concentrations of INPs relevant for mixed-phase clouds for $T>-25$℃ were measured for 82 samples collected from
May 8 to June 28. The results were linked to the abundance and properties of the observed aerosol particles (e.g., biological
particles) to examine the source of INPs and the aerosol-cloud interaction activities of different aerosol sources. Ambient
aerosols were sampled for 10 and 60 min using a high-flow rate impinger (Coriolis®micro-µ, Bertin Instruments, France, 300
L min$^{-1}$) and collected into 15 ml ultrapure water (W4502-1L, Sigma-Aldrich, US).

The offline droplet freezing assay DRINCZ (David et al., 2019) was used to measure the freezing abilities of sample droplets
at different temperatures down to −24℃. Using the sampled air and liquid volume, as well as dilution rates for preparing
DRINCZ droplets, the INP number concentration of a DRINCZ sample was calculated as a function of temperature and the
background noise was corrected by analyzing blank samples. Detailed methodology on the INP sampling protocol and
DRINCZ data analysis is provided in Gao et al. (2025).

### 2.2.2 Remote sensing instrumentation

In addition to the in situ, remote sensing instrumentation was used, such as an elastic-Laser Induced Fluorescence (LIF) lidar
and a multispectral lidar detector. An elastic-Laser Induced Fluorescence (LIF) lidar system was used to retrieve the vertical
profiles of the aerosol backscatter (b$_{aer}$) coefficients at 355 nm and the corresponding bioaerosol fluorescence backscatter
coefficients. The lidar system is based on a pulsed Nd:YAG laser (Lumibird Q-Smart 450) emitting at 355 nm with energy of
130 mJ per pulse at 10 Hz repetition frequency. The UV laser beam is emitted to the atmosphere through a Pelin-Broca prism



and a 10× diameter expander at an elevation angle of 30°, pointing northward and passing at ~50-200 m [relevant (start-end) distance of the laser beam to the surface of the mountain over a nearby hill (cf. Fig. 1-lower right). A 150 mm diameter Cassegrainian telescope (focal length f=1125 mm) was used to collect the backscattered lidar signals and fed them to a filter spectrometer through a $SiO_2$ optical fiber having a numerical angle NA=0.2 (cf. Fig. 2). At the entrance of the spectrometer a double achromatic lens (1) collimates the incoming beam, while a dichroic beam splitter (2) reflects the 355 nm lidar signal to a focal lens (7), then to a neutral density filter (6) and to the photomultiplier tube (PMT) eyepiece doublet (5). The elastic lidar signals are filtered by an interference filter (4) centred at 355 nm (FWHM=0.5 nm), while the LIF lidar signals pass through the dichroic beamsplitter and then through a cut-on long-pass optical filter (3) (FEL0400 Thorlabs GmbH) which transmits all wavelengths greater than 400 nm. An interference filter centred at 470 nm (FF02-470/100-25 Semrock, FWHM=100 nm) is used to detect the LIF lidar signals between 420-520 nm.

An optical fiber placed at the focal length of the receiving telescope is optically coupled to the entrance of a multispectral (32-channel) lidar detector (Licel GmbH) equipped with a holographic grating spectrometer (grating with 1200 grooves/mm, 400 nm blaze). The LIF lidar signals, after passing through a cut-on long-pass optical filter (FEL0400 Thorlabs GmbH), are detected in the photon counting mode by 32 photocathode elements (PMTs) and spectrally resolved between 420-620 nm with a 6.2 nm spectral resolution (Fig. 2a). In all cases, the fluorescence lidar technique is used only during nighttime due to high atmospheric background radiation during daytime. Figure 2b presents the lidar in action, while Figure 2c presents the atmospheric container-lab that housed the lidar system.





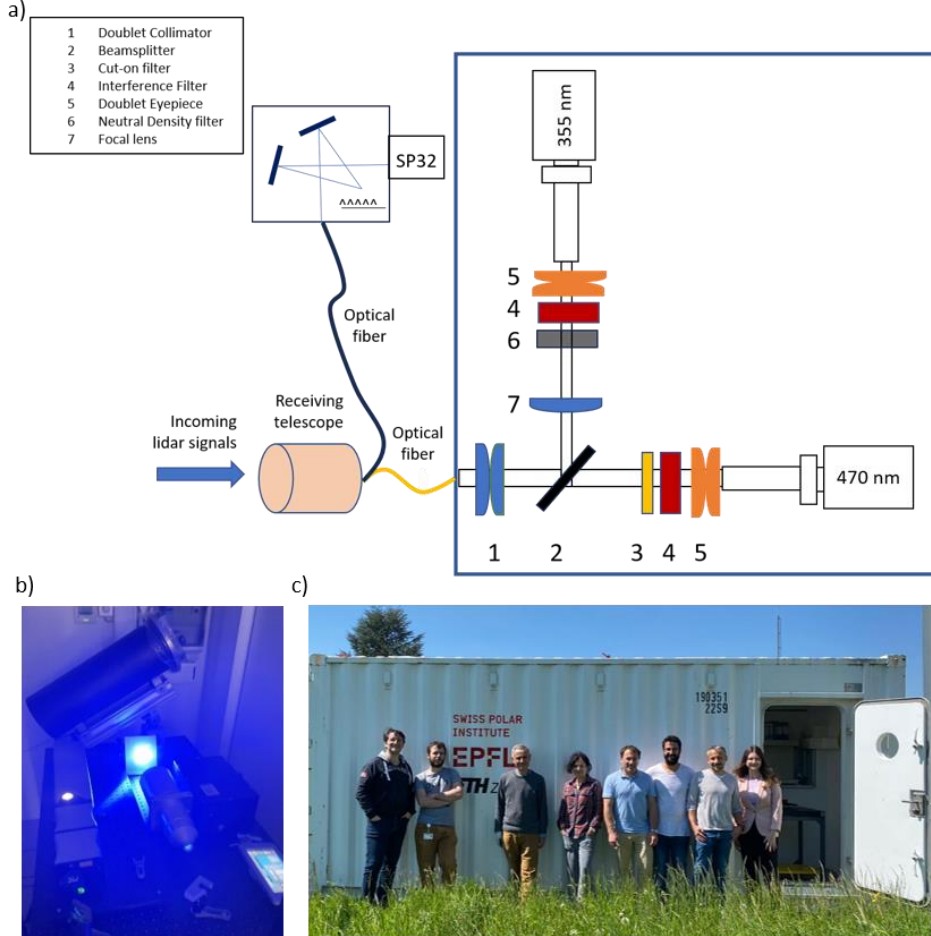

**Figure 2. (a)** Set up of the elastic-LIF lidar system and the multispectral (SP32) lidar spectrometer: (1) double achromatic lens, (2) dichroic beam splitter, (3) cut-on filter, (4) interference filter, (5) doublet eyepiece, (6) a neutral density filter and (7) focal lens **(b)** Lidar system in action **(c)** Atmospheric container-lab that housed the lidar system.

200

## 3. Methodology

### 3.1 Calibration of the fluorescence channel for the retrieval of the fluorescence backscatter coefficient

To determine the fluorescence backscatter coefficient, we need to consider the lidar signals at two channels: the elastic channel ($\lambda_E$=355 nm) and the fluorescence channel ($\lambda_F$=470 nm). Since our lidar system is not equipped with a $N_2$ Raman channel

205 (Veselovskii et al., 2020), we replace the signal of the $N_2$ Raman channel ($P_R$) by the signal at the elastic channel ($P_E$). Thus, according to Veselovskii et al. (2020) we can write the signals $P_F$ and $P_E$ as following:

$$P_F = b_F \, T_F C_F \tag{1}$$

$$P_E = b_{355} \, T_E \, C_E \tag{2}$$



where, $T_F$ and $T_E$ are the exponential terms of the atmospheric transmissions at the fluorescence and elastic channel,
respectively; $C_F$ and $C_E$ correspond to the range-independent lidar calibration constants including the efficiency of the detection
channel at wavelengths $\lambda_F$ and $\lambda_E$, respectively (Veselovskii et al., 2020). By dividing Eq. (1) by Eq. (2), the the aerosol
fluorescence backscatter coefficient $b_F$ can be obtained:

$$b_F = \frac{P_F}{P_E}\frac{T_E}{T_F}\frac{C_E}{C_F}b_{355} \tag{3}$$

To obtain the vertical profile of $b_{355}$ we used the Klett inversion technique (Klett, 1985), using typical lidar ratios at 355 nm of
45 sr for aged BB particles from North American wildfires (Hu et al., 2022; Mylonaki et al., 2021; Ortiz-Amezcua et al., 2017),
50 sr for pollen particles at low altitudes (Veselovskii et al., 2022a) and 60 sr for fresh BB particles from European wildfires
and pollen (Burton et al., 2012; Nepomuceno Pereira et al., 2014). The uncertainties of the retrieved $b_{355}$ values, are of the
order of 10-15% (Mattis et al., 2002). The molecular backscatter coefficient at 355 and 470 nm was calculated using a standard
atmosphere model by the Global Land Data Assimilation System (GLDAS) (Rodell et al., 2004); thus, the ratio $\frac{T_E}{T_F}_{|mol}$ for the
molecular contribution can be derived. As the particle atmospheric transmission differs little between 355 and 470 nm, we
used an appropriate value of the extinction-related Ångström exponent $\gamma$ equal to 1.16 (Veselovskii et al., 2020), valid for
western wildfires measured for the pair 355-532 nm (Janicka et al., 2017) to account for the spectral dependence of the aerosol
extinction between wavelengths 355 and 470 nm. As the differential particle transmission for a wavelength separation between
470 and 532 nm is less than 62 nm for low to medium aerosol loads, the induced error on $\frac{T_E}{T_F}_{|par}$ for the particulate contribution
remains less than 6% (Gast et al., 2025).

To calculate $\frac{C_E}{C_F}$ we need to consider the optical transmittances and reflectances of the various optical components (beam
splitters, interference and neutral density filters), as well as the detection sensitivities of the PMTs used at the two channels:
355 and 470 nm (Gast et al., 2025). Following Veselovskii et al. (2020), we can write:

$$\frac{C_E}{C_F} = \frac{R\ T_{ransE}}{F\ T_{ransF}}\left[\frac{P_E}{P_F}\right]_{an,cal} \tag{4}$$

where R is the reflectance of the dichroic beamsplitter (Figure 2), while $T_{ransE}$ and $T_{ransF}$ are the transmittances of the optical
elements of the LIF spectrometer at 355 and 470 nm, respectively (cf. Table 1). Using the values of the transmittance at
channels 355 and 470 nm we have: $T_{ransE} = T_1\ T_3 = 0.445$ and $T_{ransF} = T_2\ T_4 = 0,882$. Furthermore, to equalize (calibrate) the
PMT sensitivities, we installed the PMT from the fluorescence channel to the Raman one; then, by adjusting the voltage supply,
we obtained the same signal intensity $P_{F355}$ as the elastic one $P_{E355}$ at the analog channel at 355 nm  (Veselovskii et al., 2020).
This equalization (calibration) ratio can be expressed by the ratio $\left[\frac{P_{E355}}{P_{F355}}\right]_{an,cal}$ both at 355 nm, along the whole range of the
analog channel and is found to be $\left[\frac{P_{E355}}{P_{F355}}\right]_{an,cal} = 0.12$.

Furthermore, as the lidar signals $P_F$ and $P_E$ (Eq. 3) are expressed in MHz (photon counting mode) and mV (analog detection
mode) respectively, we need to introduce the factor F in equation (4), such as $F = \frac{P_{Fpc}}{P_{Fan}}$, which "equalizes" the lidar signal at



477 nm from $P_{Fpc}$ (in MHz) to $P_{Fan}$ (in mV). We note here that we selected to "equilize" the $P_{Fpc}$ (in MHz) to $P_{Fan}$ (in mV) since

the analog signals are linear in the lower altitudes in contrast to the photon counting ones which are saturated in these ranges. Thus, in our LIF lidar system we acquired a value of F=65 when gluing the $P_{Fpc}$ (in MHz) to $P_{Fan}$ (in mV). Finally, we get $\frac{c_E}{c_F}$ = 9.746 $10^{-4}$.

We also calculated the fluorescence capacity $G_F$ which characterizes the efficiency of the fluorescence with respect to the elastic scattering, defined as $G_F = \frac{b_F}{b_{355}}$ (5), is the ratio between the fluorescence and elastic aerosol backscatter coefficients.

We remind here that $G_F$ depends on the fluorescing aerosol type, as well as on the particle size and ambient relative humidity (Reichardt et al., 2018; Veselovskii et al., 2020).

**Table 1.** Transmission and reflectance of the optical elements of the LIF spectrometer at 355 nm and 470 nm channels.

| Optical elements | Elastic channel (355 nm) | Fluorescence channel (470 nm) |
|---|---|---|
| IF filters | $T_1 = 89\ \%$ | $T_2 = 98\ \%$ |
| ND filters | $T_3 = 50\ \%$ | NA |
| Beamsplitters (BS) | $R = 99,9\ \%$ | $T_4 = 90\ \%$ |

### 3.2 Fluorescence spectra clustering/analysis for pollen classification using lidar data

To classify the different taxa of pollen by using the LIF lidar technique we used the Richardson et al. (2019) approach, with improvements. The method is based on the deconvolution of the LIF lidar signals obtained by a multichannel lidar detector to determine the contribution from each taxon. For this, a database of reference fluorescence spectra (called from now on *in vivo* data) is obtained for specific pollen types prevailing at Payerne during the study period, including *Quercus robur*, *Dactylis glomerata* (proxy for grass pollen), *Fagus sylvatica* and *Betula pendula*. Those samples are representative from an optical

standpoint for the corresponding genera *Quercus*, *Fagus* and *Betula* and the grass pollen family as shown in Crouzy et al. (2016). A pulsed Nd:YAG laser (Nano L-150-10, Litron Lasers Ltd.) was used to excite the pollen samples and produce the relevant fluorescence signals, which, after passing through two long-pass optical filters (FEL0400, Thorlabs GmbH), were collected at 90° angle by an optical fiber (NA=0.22) and fed into a CCD spectrometer Ocean Optics (USB 4000 model) equipped with a 600 lines/mm  diffraction grating (spectral resolution better than 1 nm). These reference fluorescence spectra

were detected in the 400-650 nm spectral region.

As a second step, we developed a methodology to deconvolute the LIF lidar signals obtained by the multichannel lidar detector and retrieve the contributions of the relevant pollen types whose fluorescence signatures are implemented in the detected LIF lidar signals. Thus, to categorise and validate different pollen types we fitted clustering models to represent the pollen spectra during the measured periods. To this aim, we employed the Spectral Clustering algorithm (Zelnik-manor and Perona, 2004)

which is parametrized by the number of clusters, affinity and the nearest neighbours, while for the scoring function the



silhouette scorer was used (Shahapure and Nicholas, 2020). Essentially a silhouette distance is used to measure the quality of the clusters by evaluating how similar an object is to its own cluster compared to other clusters (e.g. *Quercus*, grass pollen, *Fagus* and *Betula*).

The silhouette score is calculated from the mean intra cluster distance (*in-vitro* data) and the mean nearest cluster distance (data obtained by the multichannel lidar detector) for each pollen sample (Shutaywi and Kachouie, 2021). To label our modelled clusters, we employed a similarity search between the *in-vitro* results and the fitted results. By using the centroids from our models, we computed various distances such as Euclidean, Manhattan, and Minkowski (L3 norm). Our study showed that Spectral Clustering, when paired with Euclidean and Minkowski distances, achieved the highest silhouette scores, resulting

in identifying grass pollen as the dominant pollen type, when grouped with *Quercus*, *Betula* and *Fagus*. The procedure is illustrated as a flowchart in Figure 3.

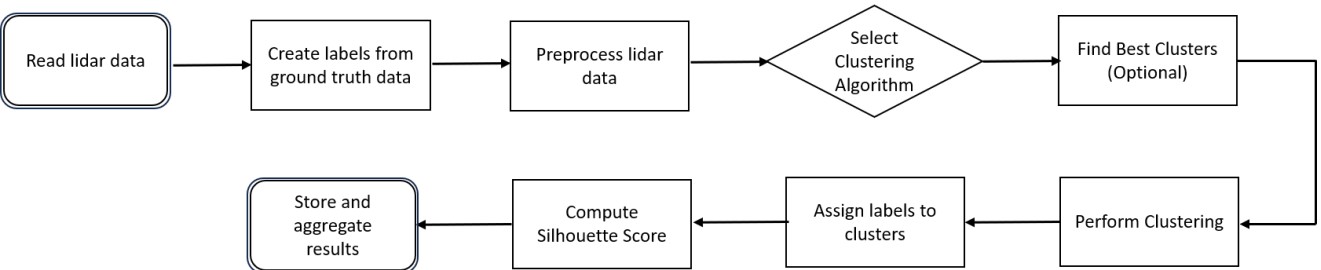

**Figure 3.** Flowchart of the classification procedure of fluorescence spectra clustering. This procedure is performed for data obtained by the
multichannel lidar detector and the in vivo fluorescence spectra.

## 4. Results

When assessing pollen levels, it is important to understand the thresholds of the exposition classes of allergenic plants (Gehrig et al., 2018; de Weger et al., 2013). These classes are categorized by low, moderate, high, and very high pollen concentrations. Even at low pollen levels, highly sensitive individuals may experience allergic symptoms. Those early manifestations can also

be related to local variations of the pollen concentration (e.g., close to sources): pollen is measured at roof level in order to increase mixing and spatial representativity. As pollen concentrations rise, both the number of affected individuals and the intensity of symptoms increase. Since pollen from different plants varies in allergenic potential, the limits for these exposure classes may also differ (Luyten et al., 2024).

Figure 4 presents a pollen calendar for the Payerne region showing the distribution of mean pollen concentrations for the 14

most significant allergenic species monitored by MeteoSwiss, as obtained by Hirst traps, for the years from 2018 to 2023. The bar colours in Figure 4 indicate the duration and the intensity of the pollen load; darker colours represent higher pollen levels, while grey color corresponds to periods when the pollen trap was out of service (October-December) and white color when concentrations were nearly zero.



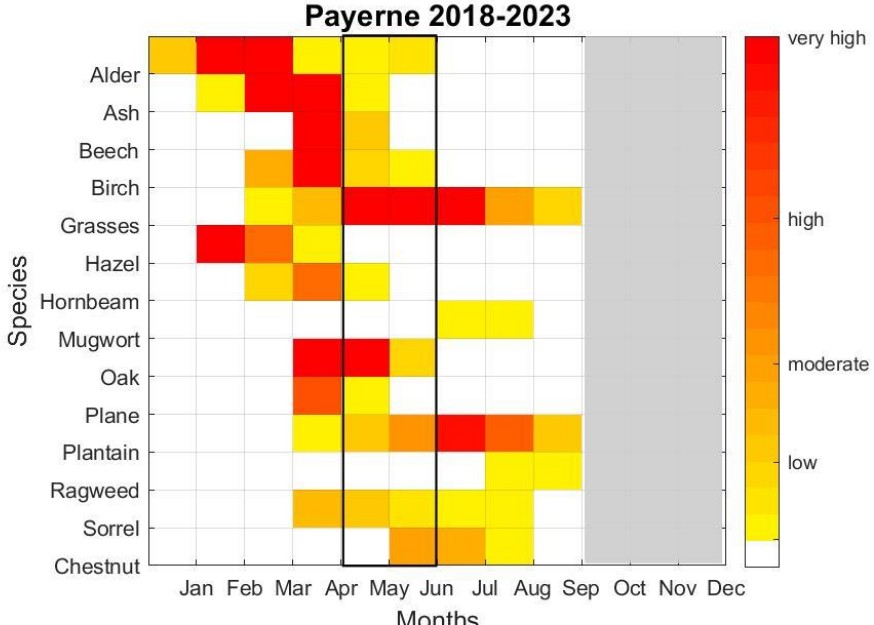

**Figure 4.** Pollen calendar for various pollen species at the Payerne monitoring station from 2018 to 2023, measured by the Hirst instrument. Gray color indicates periods when the pollen trap was out of service. Yellow, orange, red, dark red colors represent low, moderate, high, and very high pollen concentrations, respectively (adapted from Gehrig et al., 2018).

We see in this Figure that the grass pollen load can reach high to very high levels in May and June, with daily average values ranging between 5.35 to 342.76 particles m$^{-3}$ in May and from 52.13 to 271.42 particles m$^{-3}$ in June (not shown). Moreover, *Fagus* (Beech) pollen concentration is moderate during May (maximum daily value reaching 57 particles m$^{-3}$), while in June it is found to be very low. *Betula* (Birch) pollen concentrations still exhibit moderate values during May, (maximum values reaching 14.92 particles m$^{-3}$), while in June the concentrations decreased to low levels. Finally, *Quercus* (Oak) pollen presented high values in May (ranging from 0.67 to 106.45 particles m$^{-3}$) and low levels during June (values ranging from 0.82 to 38.27 particles m$^{-3}$). It is important to note that the above-mentioned concentration values represent daily averages, the instant concentration values fluctuate strongly around those averages and can reach much higher levels (Chappuis et al., 2020). In the following sections we will present two case studies of bioaerosols profiles obtained during periods of high pollination at ground level and long- and near-range transport of BB particles originating from wildfires in Canada and Germany.

## 4.1. Case 27-29 May 2023

The first case study concerns the period from 27 May (00:00 UTC) to 30 May 2023 (00:00 UTC), when the highest pollen concentrations were measured at ground level, during the PERICLES campaign. The 10-day backward trajectory analysis for the air masses arriving over Payerne at heights over 1.5 km a.s.l. in the period 27-29 May, was obtained by the HYSPLIT model (Fig. S1a,b,c), which confirmed that these air masses originated from wild forest fires (small red dots in Figure S1a,b,c





in the Supplementary section define the areas where wild forest fires were detected by the MODIS sensor) in North America and Canada, and Germany, and thus, were enriched in BB particles.

Figure 5 illustrates: (a) the temporal variation of temperature (T) and relative humidity (RH) obtained at ground level; (b) the hourly pollen concentrations obtained by SwisensPoleno (c) and Hirst (d), as well as the mass concentration of $PM_{10}$, $PM_{2.5}$, $NH_4^+$, $NO_3^-$, $SO_4^{2-}$ (e) and OC, eBC. During this period, four pollen taxa were observed: grass pollen, *Quercus, Betula*, and *Fagus*.

It is well documented that grass pollen concentration levels generally peak during daytime (generally higher temperatures
compared to nighttime ones) and often tend to increase after rainfall (Kelly et al., 2013; Sabo et al., 2015). This tendency is also evident in our case, where high temperatures and low RH values, generally coincide with elevated pollen concentrations (Fig. 5a). These observations highlight the connection between meteorological conditions (temperature, RH, precipitation, etc.) and pollen concentration and dispersion, and emphasize the role of meteorological conditions in regulating pollen concentrations near ground (Chappuis et al. 2020). It is evident from Figure 5 (b-c) that the grass pollen concentrations
consistently dominated over the other pollen types, often overpassing values of $10^3$ particles $m^{-3}$. Specifically, Fig. 5b shows maximum concentration levels of grass pollen of the order of 900-1000 particles $m^{-3}$ (mean value of 315.70 particles $m^{-3}$), while *Quercus pollen* values showed a maximum of 30 particles $m^{-3}$ (mean value of 11.24 particles $m^{-3}$). Furthermore, Hirst (Fig. 5c) detected a maximum grass pollen concentration of 2340 particles $m^{-3}$, (mean value of 639.63 particles $m^{-3}$). Moreover, the maximum values of *Quercus* were around 65 particles $m^{-3}$ (mean value of 6.76 particles $m^{-3}$). We must note here that the
mean values of *Quercus* concentrations measured by SwisensPoleno (11.24 particles $m^{-3}$) and Hirst (6.76 particles $m^{-3}$), are quite close, although the maximum values measured by Hirst are significantly higher than those measured by SwisensPoleno. Hirst also measured *Betula pollen with* maximum concentrations of the order of 33 particles $m^{-3}$ (mean value of 1.02 particles $m^{-3}$), while the *Fagus* concentrations showed a maximum value of 98 particles $m^{-3}$ (mean value of 7.46 particles $m^{-3}$). *Betula* and *Fagus* pollen were not detected by the SwisensPoleno, which could be related to the postprocessing applied to avoid false
positive detections close to detection threshold (Crouzy et al., 2022).

As the Hirst trap has a lower temporal resolution compared to the SwissensPoleno, it performs poorly at the hourly level due to limited sampling efficiency and broad band spreading, leading to significant variability in absolute concentrations across different instruments. While its performance improves at the daily level, deviations can still be observed (Adamov et al., 2024). Notably, improper calibration of Hirst flow has been documented, which may lead to deviations reaching up to 72% (Oteros
et al., 2017). These factors highlight the limitations in Hirst precision and reliability, especially for fine temporal analyses.

Furthermore, in Figure 5d we present the temporal variability of the mass concentration of $PM_{10}$, $PM_{2.5}$, $NH_4^+$, $NO_3^-$, $SO_4^{2-}$ and of OC, eBC (Fig. 5e) for the period 27 to 30 May 2023. The $PM_{10}$ levels varied from 1.9 µg $m^{-3}$ to 34.7 µg $m^{-3}$, with an average concentration of 12.162 µg $m^{-3}$. For $PM_{2.5}$, the minimum recorded value was 0.80 µg $m^{-3}$, reaching up to 18 µg $m^{-3}$, and an average of 8.30 µg $m^{-3}$. $NH_4^+$ levels ranged between 0.20 µg $m^{-3}$ and 2.67 µg $m^{-3}$, with a mean value of 0.86 µg $m^{-3}$, typical of
ambient concentrations due to agricultural activities in Payerne. Moreover, $NO_3^-$ concentrations ranged from 0.17 µg $m^{-3}$ to 6.08 µg $m^{-3}$, with an average value of 1.16 µg $m^{-3}$. Finally, $SO_4^{2-}$ concentration levels ranged between 0.96 to 4.71 µg $m^{-3}$,



with a mean value 2.53 μg m$^{-3}$. Additionally, the concentration of SO$_4^{2-}$ is influenced by both local and regional emissions (Gautam et al., 2023), however, the mean concentrations over Payerne remain quite low (Hueglin et al., 2024).



**Figure 5.** Temporal variability of **(a)** hourly temperature and RH, pollen concentration obtained by **(b)** SwisensPoleno **(c)** Hirst **(d)** of mass concentration of PM$_{10}$, PM$_{2.5}$, NH$_4^+$, NO$_3^-$, SO$_4^{2-}$ and **(e)** OC, eBC at ground level for the period 27 to 30 May 2023.

Figure 5e, shows the temporal variation of hourly averaged values of OC and eBC concentrations measured by the Aethalometer AE33 (Magee Scientific, Berkeley, CA, USA). The OC concentrations generally remained low (background values) and ranged between 3-8 μg m$^{-3}$ throughout the measurement period, with morning peaks greater than 10 μg m$^{-3}$ on 28



May (12.5 µg m$^{-3}$ at 10:00-11:00 UTC) and 29 May (19 µg m$^{-3}$ at 08:00 UTC and 12 µg m$^{-3}$ at 17:00 UTC) indicating the presence of organic particles at ground level. In contrast, eBC concentrations remained below 1 µg m$^{-3}$ and varied from 0.22 µg m$^{-3}$ to a maximum value of 1.34 µg m$^{-3}$ on 27 May at 21:00 UTC, showing no presence of BB particles at ground level.

Figures 6(a-c) present the spatio-temporal evolution of the range-corrected lidar signal (RCS) at the fluorescence channel (470 nm), while Figures 6(d-f) present the vertical profiles of the aerosol backscatter $b_{355}$ and fluorescence backscatter $b_F$

coefficients and the fluorescence capacity $G_F$. The vertical profiles of the relative humidity (RH) and potential temperature (θ) obtained by local radiosoundings at 23:00 UTC on the nights of 27-29 May 2023 are presented in Figures 6(g-i).

We note in Figure 6a (27 May) the presence of distinct nearly homogeneous aerosol layers present at the fluorescence channel (yellow and green filaments) from ground up to about 1.8 km height a.sl. (top of the PBL as derived by local radiosounding data, cf. Fig. 6g), while a thinner aerosol layer was observed between 4.7-4.9 km. Based on the backtrajectory analysis of the

air masses arriving over Payerne (at 23:00 UTC, as shown in Fig. S1a) we found that the distinct aerosol layer observed around 4.8 km was related to smoke particles released by intensive forest fires in Canada and transported over the Atlantic ocean to Switzerland. The lower yellow-green layer below 1.8 km was confined inside the PBL during the whole night. The fluorescence particles inside the PBL showed an increased fluorescence backscatter ($b_F$ ~7.4 x 10$^{-4}$ Mm$^{-1}$ sr$^{-1}$) and a well-mixed elastic backscatter coefficient ($b_{355}$ ~2.2 x 10$^{-4}$ Mm$^{-1}$ sr$^{-1}$) up to 1.8 km height (Fig. 6d). Indeed, referring to Fig. 5b,c we observed

increased values of grass pollen at ground level during (>100 particles m$^{-3}$) that night, which explains the high values of $b_F$ between 22:15 UTC (27 May) and 00:15 UTC (28 May) probably due to mixed bioaerosols (BB and pollen) and moderate fluorescence capacity indicating the presence of fluorescence particles inside the PBL. Between 2-3 km, a thin aerosol layer was identified (Fig. 6a) with moderate $b_{355}$ (1.6 Mm$^{-1}$ sr$^{-1}$) and still strong $b_F$ (<5 x 10$^{-4}$ Mm$^{-1}$ sr$^{-1}$) and increased $G_F$ (from 1.5 to 3.25 x 10$^{-4}$) values, confirming again the presence of fluorescent aerosols (Hu et al., 2022; Veselovskii et al., 2022a). Within

this layer, the RH increases with height (from 63 % to over 85 %), while θ shows a slight increase (300 K to 303 K), indicating the presence of a stable layer (cf. Fig. 6g). The thin aerosol layer around 4.5 km (Fig. 6d) exhibited increased elastic backscatter ($b_{355}$ ~1.9 Mm$^{-1}$ sr$^{-1}$) but with a lower fluorescence backscatter ($b_F$ ~1.4 x 10$^{-4}$ Mm$^{-1}$ sr$^{-1}$) coefficient, indicating fewer fluorescence aerosols due to long-range transport of dried BB aerosols (Reichardt et al., 2025), as the RH decreased down to ~8%.





**Figure 6. (a-c)** Spatio-temporal evolution of the range-corrected lidar signal (RCS) at the fluorescence channel (470 nm) **(d-f)** aerosol $b_{355}$ and fluorescence $b_F$ backscatter coefficients and $G_F$ **(g-i)** RH and $\theta$ as obtained by radiosonde launches for the period 27-29 May 2023.

On the following day 28 May, aerosols are more dispersed above the PBL (located ~2 km a.sl.), with a ~1.8-km thick aerosol layer extending from 2.2 to 4.0 km (Fig. 6b). The aerosols inside a quite well-mixed PBL, exhibited similar trends as during the previous night, with slightly lower $b_{355}$ values ($b_{355}$~1.4 Mm$^{-1}$ sr$^{-1}$), but still with quite high $b_F$ values ($b_F$ ~3.8 x 10$^{-4}$ Mm$^{-1}$ sr$^{-1}$) and notably increased $G_F$ values ($G_F$~4 x 10$^{-4}$) at the top of the PBL, indicative of the presence of fluorescent aerosols (Hu et al., 2022; Veselovskii et al., 2022a,b) inside the PBL (Figs. 6e,h), linked again to high pollen concentrations of grass pollen





(>100  particles m$^{-3}$) (Fig. 5b,c) at ground. In the aerosol layer between 2.0-3.5 km (Fig. 6e between 22:00-23:00 UTC), b$_{355}$ increased to 1.6-2.0 Mm$^{-1}$ sr$^{-1}$, while b$_F$ and G$_F$ decreased to 2.3 x 10$^{-4}$ Mm$^{-1}$ sr$^{-1}$ and 1.1 x 10$^{-4}$, respectively, showing a weaker

fluorescence signal than inside the PBL, probably due to mixed BB with continental polluted aerosols. The RH values increased inside that layer (from 24 % to 52 %), while θ seem to be constant (Fig. 6h). Between 3.5-5.0 km, we observed a similar pattern as in the second aerosol layer regarding the optical properties, while the RH seem to be stable at around 14 % and θ slightly increases from 305 K to 308 K (Fig. 6h) in this stable layer.  As on the previous day, the air masses arriving between 2.0 and 5.0 km height are related to long-range transport of BB aerosols, showing intense fluorescence (Reichardt et al., 2025).

On 29 May, the spatio-temporal evolution of the RCS signals at 470 nm, shows a quite well-mixed fluorescence layer from 1.5 to 3.5 km height (b$_F$~2 x 10$^{-4}$ Mm$^{-1}$ sr$^{-1}$), with increased b$_F$ values (b$_F$~3 x 10$^{-4}$ Mm$^{-1}$ sr$^{-1}$) over 3.5 km (Fig. 6c). On that day the PBL height was confined below 1.2 km, where a distinct thin fluorescence layer is observed (Fig. 6c) with b$_F$~1.5 x 10$^{-4}$ Mm$^{-1}$ sr$^{-1}$ during the first observation period (00:50 – 01:50 UTC), with similar value in the next hour (01:50 – 02:50 UTC) (Fig. 6i). This is again linked to the presence of fluorescent pollen aerosols of grass pollen (>100 particles m$^{-3}$) at ground as

shown in Fig. 5b,c. In the free troposphere just above the PBL (~1.5 km height) a second thin fluorescence layer (~ 2 km height) is observed (Fig. 6c). Meanwhile, the G$_F$ values show a nearly constant profile from the top of the PBL to ~3.5 km (G$_F$~2.7 x 10$^{-4}$ between 00:50 – 01:50 UTC (Fig. 6f) and G$_F$ ~1.5 x 10$^{-4}$ between 01:50 – 02:50 UTC (Fig. 6i), with enhanced G$_F$ values between 3.5 and 4.5 km (3.0 - 4.2 x 10$^{-4}$ between 00:50 – 01:50 UTC (Figs. 6e,f) and ~2.8 x 10$^{-4}$ (01:50 – 02:50 UTC). We observe that the RH (Fig. 6i) varied slowly from 68 % to 47 % (2.0-4.0 km height), with approximately stable

aerosol optical properties (b$_{aer}$, G$_F$, b$_F$), within a stable atmospheric layer (θ ranged from 300 K to 306 K) (Fig 6i). These observations during May, highlighted the presence of fresh (<2 days) and aged (>5 days) fluorescence BB particles in the altitude region 1.5-5.0 km heigh a.s.l. over Payerne, transported from the wildfires in Germany and Canada, respectively (Fig. S1a,b,c).

As a next step we deconvoluted the fluorescence spectra obtained by the lidar multichannel spectrometer over 1-hour period

and averaged in the height region from ground up to 1.2 km a.s.l. (Fig. 7a) into the pollen types measured at ground level by SwisensPoleno and the Hirst trap, to examine the feasibility of the lidar multichannel spectrometer LIF technique to retrieve and categorise the different pollen types aloft. The validation of our methodology will refer to the ground pollen data with increased concentrations during the observation period from 27 to 29 May, *Fagus*, *Betula* and *Quercus*, with the grass pollen being the dominating pollen taxa (Fig. 5b,c). To this end, at first Figure 7a presents the *in vivo* LIF fluorescence spectra (dashed

lines) of four different pollen types -*Dactylis glomerata* (proxy for grass pollen), *Fagus sylvatica*, *Betula pendula* and *Quercus robur* obtained by laser excitation at 355 nm under laboratory conditions (Richardson et al., 2019). Additionally, in the same figure we present the fluorescence spectra measured by the lidar multichannel spectrometer during the observation period from 27 to 29 May, averaged in the height region from ground up to 1.2 km a.s.l height (continuous lines). Then, we followed the methodology explained in Section 3.2 applied to lidar multichannel spectrometer signals and the corresponding *in vivo* spectra

of these selected pollen types. In Figure 7b we quantify the presence of each pollen type aloft as a percentage of pollen counts for each day of the observed period. On 27 May, grass pollen was the most prevalent pollen type in the atmosphere, contributing

 

to ~ 87% in the lidar multichannel spectrometer signal, while *Quercus* accounted for ~13%. These results are in accordance with the *in-situ* data obtained from SwisensPoleno and Hirst (Fig. S2), where grass pollen represented 97% and *Quercus* accounted for 3%. On 28 May, the presence of grass pollen reached 80%, with *Betula, Fagus* and *Quercus* reaching 8%, 6%

and 5%, respectively. In this case, the *in-situ* data similarly indicated an increased percentage of grass pollen at approximately 95%, with *Quercus* and *Fagus* contributing 4% and 1%, respectively, while *Fagus* concentrations were generally low or absent. On May 29, the algorithm detected both grass pollen reaching a peak of 85%, and *Betula* at 15% (Fig. 7b), in accordance with the Hirst and SwisensPoleno data, which indicated high concentration (~250-300 particles m$^{-3}$ at 23:00 UTC) of grass pollen along with a small presence of *Quercus* (~10 particles m$^{-3}$) concentrations, while *Betula* was absent at that day.

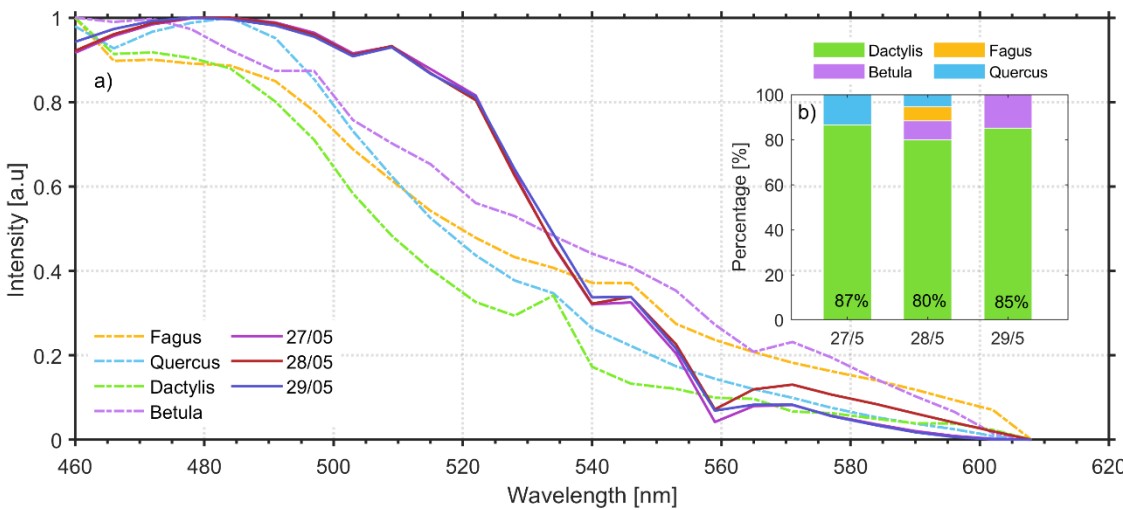


**Figure 7. (a)** Pollen fluorescence spectra of *Fagus, Quercus, Dactylis* and *Betula* obtained *in vivo* by lidar excitation at 355 nm (dashed lines) and pollen spectra (continuous lines) obtained by the lidar multichannel spectrometer, along with the **(b)** percentages of each pollen type for various clustering algorithms and metrics (from 27-29 May) from ground up to 1.2 km a.s.l.

**4.2. Case 12-15 June 2023**

Figure 8a presents the daily evolution of temperature and RH at ground level. The pattern observed during the first case study (27-29 May 2023) is evident here as well, with increased pollen concentrations coinciding with the highest temperature and lowest RH values. Regarding the temporal variations of the hourly pollen concentrations measured at ground level, the SwisensPoleno did not detect any pollen concentration of *Betula, Quercus* and *Fagus* as during May 2023, while grass pollen was recorded at a maximum concentration of 644 particles m$^{-3}$ and a mean one of 159.81 particles m$^{-3}$ (Fig. 8b). Hirst measured

a mean concentration of grass pollen equal to 196.47 particles m$^{-3}$ with a maximum of 878 particles m$^{-3}$, while *Quercus, Fagus* and *Betula* showed very low concentrations (around 15 particles m$^{-3}$) (Fig. 8c). Furthermore, the PM$_{10}$ levels varied from 7.6 μg m$^{-3}$ to 23.3 μg m$^{-3}$, with an average concentration of 15.6 μg m$^{-3}$. For PM$_{2.5}$, the minimum recorded value was 5.5 μg m$^{-3}$, reaching up to 13.1 μg m$^{-3}$, and an average of 7.40 μg m$^{-3}$. (Fig. 8d). Moreover, the OC concentrations generally ranged



between 2.33-5.40 µg m$^{-3}$, while the eBC ones were again very low with values ranging from 0.08 to 1.25 µg m$^{-3}$ (Fig. 8e),

indicating the absence of BB aerosols.



**Figure 8. (a)** Temporal evolution of daily temperature and RH from MeteoSwiss data, as well as, evolution of the hourly pollen concentration obtained by **(b)** SwisensPoleno **(c)** Hirst **(d)** of mass concentration of PM$_{10}$, PM$_{2.5}$, and **(e)** OC, eBC at ground level, **(f)** WIBS$_{ABC}$, WIBS$_{coarse}$, and WIBS$_{total}$ particle number concentrations **(g)** INP concentrations at −14℃, −15℃ and −20℃, and the correlations between **(h)** INPs at

−14ºC and WIBS$_{ABC}$ **(i)** INPs at −20ºC and WIBS$_{ABC}$ for the period between 12 and 15 June 2023.



Figure 8f shows the number concentrations of bioaerosols measured at the WIBS$_{total}$, WIBS$_{coarse}$, and WIBS$_{ABC}$ channels. The WIBS$_{ABC}$ channel measured concentrations between 1000 and 10000 particles cm$^{-3}$, which is at least two orders of magnitude than the pollen number concentrations shown in Figures 8b to c. This is because the WIBS$_{ABC}$ channel refers to the presence of a wide range of fluorescent biological particles, including pollen particles. Daily minimum WIBS$_{ABC}$ number concentrations

tend to appear in the morning hours, whereas maximum values show up after midnight, more specifically on June 12 to 14 (Fig. 8f). Fig. 8g presents the INP number concentrations at –14 and –20℃.

The INP concentrations at –20℃ are higher than those at –14℃ by more than one order of magnitude. To investigate the sources of INPs, we compared the correlations between the INP number concentrations and the WIBS$_{total}$, WIBS$_{coarse}$, and WIBS$_{ABC}$ ones, as well as the total pollen number concentrations measured by the SwissensPoleno, and Hirst, and the mass

concentrations of eBC and OC. We found that the INPs activated at –14 and –20℃ show insignificant correlations with OC (Fig. S3a, b) and eBC (Fig. S3c, d), thus, suggesting a relative unimportant role of BB particles on mixed-phase cloud formation (Gao et al., 2025). In contrast, the WIBS$_{total}$ and INPs are positively correlated, but with a low correlation coefficient (Figure S4a, b). This may be due to a minor fraction of INPs in the total number of aerosol particles (Fig. 8f,g). More precisely, the INPs at –14℃ have significant correlations with WIBS$_{ABC}$ particles (Fig. 8h), two of which show $\rho_{Pearson}$= 0.64 ($p_{pearson}$=0.03)

and $\rho_{Spearman}$= 0.64 ($p_{Spearman}$=0.04). This indicates the important contribution of FBAPs in INPs activated at –14℃. However, the correlation analysis shows that the detected pollen particle number concentrations from SwisensPoleno (Figure S5a) and Hirst (Fig. S5c) are insignificantly correlated with INPs at –14℃. This may indicate that not all types of pollen particles are active INPs and the INPs originated from FBAPs may include a broad range of biological particles in addition to pollen (Gao et al., 2024; Kanji et al., 2017; Morris et al., 2013). In addition, the INPs at –20℃ are correlated with WIBS$_{coarse}$ particles (Fig.

S4c, d), by showing $\rho_{Pearson}$= 0.72 ($p_{pearson}$=0.01) and $\rho_{Spearman}$= 0.52 ($p_{Spearman}$=0.10), which suggests coarse-sized dust particles may contribute to the observed INPs at –20℃. The above results are consistent with similar finding presented in the literature and confirm that the biological particles are more relevant INPs for $T > -15$℃, while dust particles generally contribute cloud ice formation for $T < -15$ °C (Murray et al., 2012).

Figures 9(a-d) present the spatio-temporal evolution of the range-corrected lidar signal (RCS) at the fluorescence channel (470

nm), while Figures 9(e-h) present the vertical profiles of b$_{355}$, b$_F$ and G$_F$. The vertical profiles of RH and θ obtained by local radiosondings at 23:00 UTC on the nights 12-15 June 2023, are presented in Figures 9(i-l). The general overview of Figs. 9(a-d) shows the presence of aerosol fluorescent layers from ground up to 2-2.5 km heigh, which at near ground level (below 1.2-1.5 km) may be influenced mainly by the high concentrations of grass pollen from local sources, as discussed in Figs. 8b,c. The backward trajectory analysis of the air masses arriving over Payerne between 1.5-4.0 km during the nights of 12-15 June

indicated as aerosol source region the wildfires occurring in Germany (Fig. S9a,b,c,d).





**Figure 9. (a-d)** Spatio-temporal evolution of the range-corrected lidar signal (RCS) at the fluorescence channel (470 nm) **(d-g)** aerosol $b_{355}$ and fluorescence $b_F$ backscatter coefficients and $G_F$ **(h-j)** RH and θ as obtained by radiosonde launches for the period 12-15 June 2023.





More specifically, on 12 June nearly homogeneous aerosol RCS signals at the fluorescence channel are observed inside the
PBL (from ground up to 1.8 km) (Fig. 9a), as also shown in Fig. 9e on the same heights, at the vertical profiles of $b_{355}$, $b_F$ and
$G_F$. This homogeneous layer is characterized by $b_F$ values ranging between $4.5 \times 10^{-4}$ $Mm^{-1}$ $sr^{-1}$ (20:50-21:50 UTC) and $5.2 \times 10^{-4}$ $Mm^{-1}$ $sr^{-1}$ (23:00-00:00 UTC), and $G_F$ ones between $3.2 \times 10^{-4}$ (20:50-21:50 UTC) and $4.5 \times 10^{-4}$ (23:00-00:00 UTC). The
vertical profile of $b_{355}$ in the same layer shows moderate values of the order of 1.4 $Mm^{-1}$ $sr^{-1}$ during that night (Fig. 9e). The
RH values are close to 50% up to 1.8 km height indicating a nearly well-mixed and stable ($d\theta/dz>0$) PBL layer. At heights
from 2.5 to 3.5 km, where again similar stable meteorological conditions prevail (atmospheric stability, since $d\theta/dz>0$) and
RH~40-50%), the $G_F$ shows increased values of $7.3 \times 10^{-4}$ (20:50-21:50 UTC) and even up to $10^{-5}$ (23:00-00:00 UTC), while
the $b_{355}$ (~1.0 $Mm^{-1}$ $sr^{-1}$) and $b_F$ (~$3 \times 10^{-4}$ $Mm^{-1}$ $sr^{-1}$) profiles show nearly constant values during that night. At the top (at 3.5
km) of this 1-km thick atmospheric layer very humid air masses are observed (RH~85%) giving rise to a local peak of $b_{355}$
indicating a possible particle hygroscopic growth (Miri et al., 2024; Navas-Guzmán et al., 2019). Over 3.5 km height thin
filaments of smoke particles are observed around 3.8 and 4.4 km (Fig 9a), while the relevant $b_{355}$ and $b_F$ profiles present
decreasing trends (Fig. 9e).

On 13 June, again nearly homogeneous aerosol RCS signals at the fluorescence channel were observed inside the PBL (from
ground up to 1.8 km) (Fig. 9b) with a fluorescing smoke filament observed at 2.2 km. The $b_F$ profile is nearly constant
($b_F$~$4.2 \times 10^{-4}$ $Mm^{-1}$ $sr^{-1}$) with height up to 3.8 km, except between 1.5 and 2.2 km where it presents a local maximum of 4.2-
6.2 $Mm^{-1}$ $sr^{-1}$ (20:50-21:50 UTC). Later that night (23:00-23:30 UTC) it becomes nearly constant (~4.5-5.0 $Mm^{-1}$ $sr^{-1}$) with
height from ground up to 3.8 km (Fig. 9f). The $G_F$ profiles are nearly homogeneous up to 3.8 km, except the height region
between 2.2-2.5 up to 2.9 km where they show pronounced maxima due to very low $b_F$ values. The profile of $d\theta/dz>0$ remains
positive, indicating stable meteorological conditions up to 3.8 km.

On 14 June, again nearly homogeneous aerosol RCS signals at the fluorescence channel were observed inside the PBL (from
ground up to 1.9 km) with decreasing values up to 2-2.7 km (Fig. 9c). The vertical profiles of $b_{355}$ (mean value $2.0 \times 10^{-4}$ $Mm^{-1}$ $sr^{-1}$) and $b_F$ (mean value $6 \times 10^{-4}$ $Mm^{-1}$ $sr^{-1}$) show a well-mixed layer up to 1.9-2.0 km height with increasing values up to 2.7
km. The $G_F$ profiles (mean values of $3 \times 10^{-4}$) are nearly constant up to 2.0 km with slightly decreasing values up to 2.7 km.
Over that height the $b_{355}$ and $b_F$ profiles show decreasing values indicative of the presence of low aerosol concentrations (Fig.
9g). In Fig. 9k we note the presence of a humid layer (60%<RH<85%) between 2.0-3.0 km which leads to an increase of $b_{355}$
and $b_F$ values around 2.7 km possibly due to hygroscopicity growth and enhanced fluorescence, respectively (Miri et al., 2014).

On June 15, more intense than last days' aerosol RCS signals at the fluorescence channel were observed, below 1.6 km,
especially during the late evening hours (23:30-00:30 UTC), with decreasing intensity up to ~2.5 km (Fig. 9d). From 22:30 to
23:30 UTC the vertical profile of $b_{355}$ (mean value $1.2 \times 10^{-4}$ $Mm^{-1}$ $sr^{-1}$) is homogeneously distributed up to 2.2 km with
increasing values up to 3.0 km (due to the presence of a humid layer with RH of 85%) and then decreasing values up to 5 km.
The $b_F$ profile (mean value $3.5 \times 10^{-4}$ $Mm^{-1}$ $sr^{-1}$) shows a very well-mixed layer from ground up to 3.0 km height, with
decreasing values at higher altitudes. The $G_F$ profile is nearly constant up to 2.3 km (mean values ~$2.9 \times 10^{-4}$) and decreases
up to 5 km. Similar behavior show the vertical profiles of $b_F$ and $G_F$ from ground up to 3.1 km, in the period 23:30-00:30 UTC





except the $b_{355}$ profile which remains continuously well distributed below that height. On this day the decreasing values of the $b_{355}$, $b_F$ and $G_F$ profiles over 3.1 km height are indicative of the presence of low bioaerosol concentrations.

Overall, in most studied cases in June, we found that the enhanced values of $b_{355}$ and $b_F$ below 2-3 km down to 1.5 k are linked to increased fluorescence backscatter attributed to BB particles originating from wildfires in Germany. However, the enhanced values of the aerosol RCS signals at the fluorescence channel below 1.5 km are due to the presence of high concentrations of grass pollen as also found near ground (Fig. 8b,c).

As in the previous case study, we deconvoluted the fluorescence spectra obtained by the lidar multichannel spectrometer over
1-hour period and averaged in the height region from ground up to 1.2 km a.s.l. for the period 12-15 June (Fig. 10a) into the pollen types measured at ground level by SwisensPoleno and the Hirst trap, to retrieve and categorise the different pollen types aloft.

Figure 10a presents the *in vivo* LIF fluorescence spectra (dashed lines) of four different pollen types -*Dactylis glomerata* (proxy for grass pollen), *Fagus sylvatica*, *Betula pendula* and *Quercus robur*- obtained by laser excitation at 355 nm under
laboratory conditions as explained in the previous case of May 2023. Additionally, in the same figure we present the fluorescence spectra obtained by the lidar multichannel spectrometer from 12 to15 June, averaged in the height region from ground up to 1.2 km a.s.l height (continuous lines).

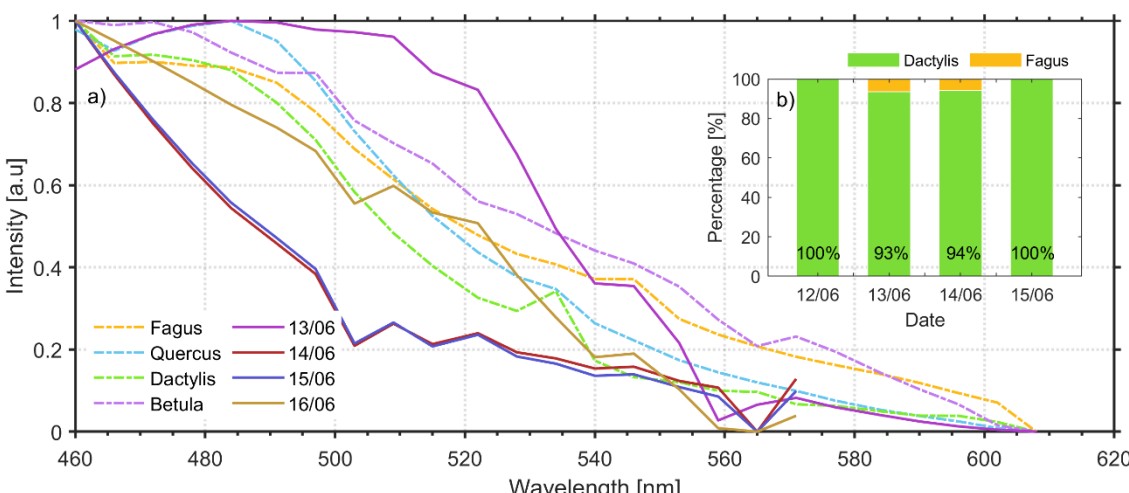

**Figure 10. (a)** Pollen fluorescence spectra of *Fagus, Quercus, Dactylis* and *Betula* obtained *in vivo* by lidar excitation at 355 nm (dashed
lines) and pollen spectra (continuous lines) obtained by the lidar multichannel spectrometer, along with **(b)** the percentages of each pollen type for various clustering algorithms and metrics (from 12-15 June) from ground up to 1.2 km a.s.l.

In Figure 10b we quantify the presence of each pollen type aloft as a percentage for each day of the observed period. Based on our algorithm, we observed that on most days, grass pollen was the only pollen type detected. For instance, on 12 June, grass pollen showed a 100% presence, while on 13 and 14 June, *Fagus sylvatica* was also detected, though only with 7% and 6%
presence, respectively. On 15 June, grass pollen once again dominated with 100% presence in the atmosphere. These results are in good agreement with the *in-situ* pollen measurements from SwisensPoleno and Hirst (Fig. S7), confirming that grass





pollen was the most dominant pollen type in the atmosphere. Regarding the *in-situ* measurements (Fig. S7), grass pollen again showed the highest concentrations along all days, with peaks around 79% on 12 and 14 June. *Betula* exhibited moderate concentrations, reaching highest percentages of ~33% on 15 June. This is probably related to the uncertainties of the method,

as birch trees are not typically flowering during that period. *Quercus* and *Fagus* showed very low or zero concentrations, with *Quercus* peaking at around 7% on 13 June, while *Fagus* presence remaining absent throughout the period. The *in-situ* data highlight again the predominance of grass pollenover a noticeable presence of *Betula* and near-zero concentrations of all other pollen types.

## 5. Discussion and Conclusions

In the two-case studies analyzed in the preceding sections we focused on the detection of bioparticles in the first 5 km a.s.l. and found that from near ground to ~1.0-1.5 km height, pollen particles dominate, while over 1.8-2.0 km smoke layers or distinct filaments of BB particles are observed. The smoke particles were found to originate from long- and/or near-range wild forest fires in Canada and Germany occurred during May and June 2023 (Figs. S1 and S6, respectively). In this section we will examine how the aging of the smoke particles from the Canadian forest fires affects the $b_F$ values aloft, embedded in the

same air masses overpassing a central European lidar station (at heights below 8 km) before reaching Payerne. To this end we will use published data of $b_F$ from the literature at 532 nm from Lindenberg, Germany (DWD), (Reichardt et al., 2025), and Payerne, Switzerland (PAY) (this study) from 26 to 29 May 2023.

Figure 12a shows the $b_F$ profiles over DWD (26 May, 21:00-2200 UTC) and PAY (28 May, 21:00-23:00 UTC). In this figure we show with red line the air mass trajectories which overpassed the DWD station at 3.5 km (26 May, 21:00-22:00 UTC), then

arrived two days later over PAY (28 May, 21:00-23:00 UTC) around 3.0 km height. Along this trajectory the $b_F$ values starting over DWD at 3.5 km they presented quite high values (~5.0 x $10^{-4}$ Mm$^{-1}$sr$^{-1}$), but during their descent to lower heights (3.0 km) on the following days (27-28 May) they were characterized by lower values by a factor ~3 (~1.8 x $10^{-4}$ Mm$^{-1}$sr$^{-1}$) indicating smoke aerosols with lower fluorescence potential, probably due to mixing with non-fluorescent (e.g. continental polluted) aerosols of different RH values at lower heights. Similar behavior showed the air masses (blue line) in Fig. 12b, starting at 4.5

km over DWD (26 May, 21:00-22:00 UTC), they reached PAY on 30 May (00:50–02:50 UTC) at 2.5 km. Along this transport the $b_F$ values passed from ~4.5 x $10^{-4}$ Mm$^{-1}$sr$^{-1}$ to 2.0 x $10^{-4}$ Mm$^{-1}$sr$^{-1}$, now showing a ~50% reduction on the $b_F$ values, indicating again a possible mixing with non-fluorescent (e.g. continental polluted) aerosols of different RH values at lower heights.




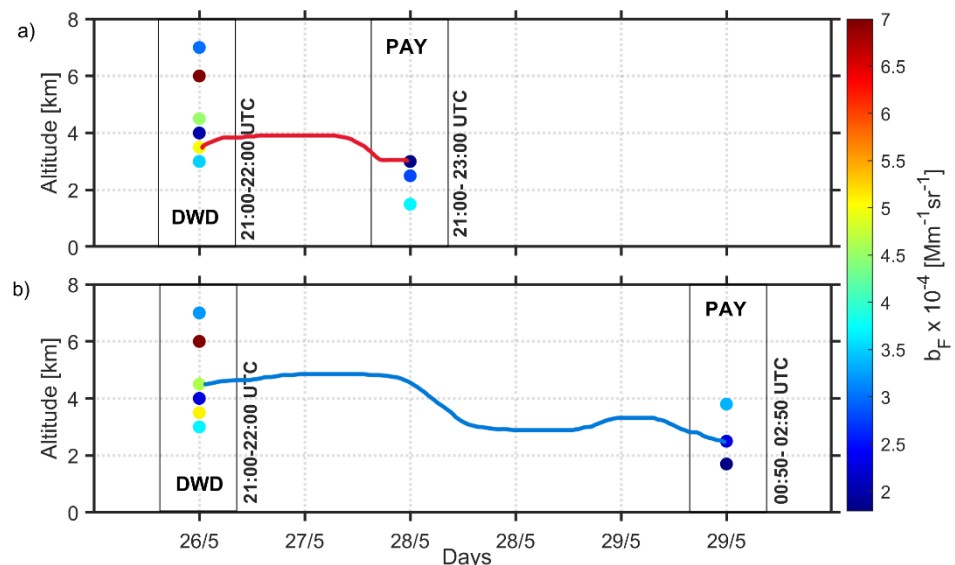

**Figure 11.** Temporal and vertical variations of $b_F$ measured over two stations (DWD and PAY) between 1.8 and 8 km height, during the BB aerosols from wild forest fires from **(a)** long-range transport from Canada (26 to 28 May 2023) **(b)** near-range transport from Germany (26 to 29 May 2023). The observing time periods are indicated also for each station.

During the PERICLES campaign we extended the LIF lidar technique using a single-channel fluorescence lidar system coupled with a 32-channel spectrometer to classify pollen types near ground (from ground up to 1.2 km a.s.l.) and to characterize atmospheric layers of pollen and near- and long-range BB particles in the free troposphere up to 5 km height, combining the experimental setups implemented by Richardson et al. (2019) and Veselovskii et al. (2021). The measurements were performed over Payerne during the spring-summer wildfire and intense pollination season on May-June 2023. The collected pollen data

near ground using Hirst and SwisensPoleno samplers were compared to multi-channel LIF lidar data used to derive the percentage of pollen counts of *Quercus*, grass pollen, *Fagus* and *Betula* over 200 m from ground. Air pollution *in situ* data ($PM_{10}$, $PM_{2.5}$, $NH_4^+$, $Cl^-$, $NO_3^-$, $SO_4^{-2}$, eBC, OC) were used to characterize the air masses sampled near ground as contaminated or not by air masses aloft containing BB aerosols. INPs measurements were also performed to evaluate their role of bioaerosols in cloud formation.

During PERICLES, the LIF lidar measurements revealed well-mixed fluorescence aerosol layers due to high concentrations of pollen inside the PBL, or in the adjunct layer (2-3 km height), as these layers exhibited generally strong fluorescence backscatter ($1.5 \times 10^{-4}$ to $7.5 \times 10^{-4}$ $Mm^{-1}sr^{-1}$) and well-mixed elastic backscatter coefficients. In contrast, higher-altitude layers (3-5 km) showed, generally, weaker $b_F$ values ($< 4 \times 10^{-4}$ $Mm^{-1}sr^{-1}$) in the case of long-range transport of aged smoke particles (Canadian forest fires), in contrast to higher values ($3.0 \times 10^{-4}$ $Mm^{-1}sr^{-1}$ to $7.5 \times 10^{-4}$ $Mm^{-1}sr^{-1}$) from near-range fresh smoke

particles (German forest fires), indicating the crucial role of aging of the smoke particles on the $b_F$ coefficients. We also note here the role of RH on the increase (mainly below 2-3 km height) and becomes more relevant for the fresh German smoke particles (12-14 June 2023) against a general decrease of the $b_F$ values with higher RH values for the aged Canadian smoke particles. The fluorescence capacity $G_F$ inside the smoke layers ranged between $1.5 \times 10^{-4} - 4.2 \times 10^{-4}$, while that of pollen



particles varied between ~ $1.0 \times 10^{-4} – 4.5 \times 10^{-4}$ (inside the PBL), aligning with values reported in similar cases by Gast et al.( 2025), Hu et al. (2022), Reichardt et al. (2024) and Veselovskii et al. (2022b). Our observations highlight the importance of the age (long- or near-range transport) of the smoke particles in relation with the prevailing RH values aloft.

We also revealed the strong influence of meteorological conditions (RH and temperatures values) on pollen concentration near ground. In all days, pollen of grass pollen consistently dominated all other pollen types, by a significant margin.

Furthermore, the number concentrations of INPs relevant for mixed-phase clouds for temperatures between -14℃ and -25℃ were also measured. The INP concentrations at –20°C were found to be more than an order of magnitude higher than those at –14°C, which highlights a strong temperature dependence and consistent with literature (Kanji et al., 2017; Murray et al., 2012). Our correlation analysis revealed that INPs activated at -14℃ and -25℃ had the lowest correlations with eBC and OC concentrations, suggesting that BB particles play a minor role in mixed-phase cloud formation. However, significant correlations were observed between INPs at –14°C and WIBS$_{ABC}$ particles, indicating a notable contribution from FBAPs, while pollen concentrations showed insignificant correlations.

Our novel methodology for characterizing pollen types by deconvoluting LIF lidar signals and comparing them with reference fluorescence spectra from an extensive pollen database. Validation with *in-situ* measurements confirmed that grass pollenwas identified as the predominant pollen type, with smaller contributions from *Fagus*, *Betula*, and *Quercus*. These findings highlight the effectiveness of LIF lidar techniques for classifying pollen types aloft, especially in parts of the season where few taxa provide the main contribution to the total pollen concentration.

The analysis of b$_F$ values across four European LIF lidar stations revealed that aged air masses containing smoke particles can show a ~50% reduction of their b$_F$ values during their transport in the free troposphere (3-5 km) due to mixing with other non-fluorescent particles with different hygroscopicity potential. Moreover, our study confirmed the capability of LIF lidar techniques equipped with elastic and advanced fluorescence detection capabilities (32-channels) to distinguish different types of pollen in conjunction with the detection of fluorescing smoke in free tropospheric atmospheric layers. Furthermore, our findings contribute to a more comprehensive understanding of the role of bioaerosols in cloud formation and climate dynamics and thus, highlighting the need for systematic LIF lidar high spectral resolution measurements in conjunction with complete bioaerosols characterization near ground.

*Author contributions.* AN, BCr and AP organized the PERICLES campaign. MG and AP conceived and led this study. MG and AP developed the optical elements of the LIF lidar. AP, KG, SE, AN and EG conducted the experiments and collected the raw data. CZ and ZAK provided the DRINCZ data. BCr, SE and GL provided the in situ pollen data. BCl and BCa supported the campaign at MeteoSwiss Payerne. MCC provided the air quality data. MG analysed the lidar data and interpreted the results with inputs from AP, RF, KG, PG and BCr. MG and AP wrote the original manuscript with input from KG, AN, BCr, PG. MG prepared the figures with contributions from AP, KG, PG and BCr. RF wrote the analysis software for the lidar fluorescence date. PG performed the clustering algorithms. BS provided the pollen samples. All authors discussed, reviewed, and edited the manuscript.



*Acknowledgements.* The authors acknowledge the NOAA Air Resources Laboratory (ARL) for the provision of the HYSPLIT transport and dispersion model and READY website (https://www.ready.noaa.gov). The authors also acknowledge the use of
data from NASA's Fire Information for Resource Management System (FIRMS) (https://www.earthdata.nasa.gov/data/tools/firms), part of NASA's Earth Science Data and Information System (ESDIS). We acknowledge and thank the technical team from EPFL, MeteoSwiss in Payerne and ETHZ, Switzerland for the experimental support. The maps shown in Fig.1 are provided by Google Maps. We also acknowledge the technical support of Kostas Hormovas (Physics-NTUA).


*Financial support.* This research has been supported by the European Research Council PyroTRACH project (project ID 726165) funded from H2020-EU.1.1. (ERC), the Swiss National Science Foundation project 192292, Atmospheric Acidity Interactions with Dust and its Impacts (AAIDI) and the Horizon Europe CleanCloud project (Grant agreement No. 101137639). M.G. was supported by the Hellenic Foundation for Research and Innovation (HFRI) under the 4[th] Call for HFRI Ph.D.
Fellowships (Fellowship number: 9293).

*Competing interests.* The authors declare that they have no conflict of interest.

*Data availability.* Lidar, Hirst, SwisensPoleno, air pollution and radiosounding data are available upon request from the
corresponding author (alexandros.papagiannis@epfl.ch). The air mass backward trajectory analysis is based on air mass transport computation by the NOAA (National Oceanic and Atmospheric Administration) HYSPLIT (HYbrid Single-Particle Lagrangian Integrated Trajectory) model (http://ready.arl.noaa.gov/HYSPLIT_traj.php). GDAS1 (Global Data Assimilation System 1) re-analysis products from the National Weather Service's National Centers for Environmental Prediction are available at https://www.ready.noaa.gov/ gdas1.php.

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
