# Peer review of "Profiling pollen and biomass burning particles over Payerne, Switzerland using laser-induced fluorescence lidar and *in situ* techniques during the 2023 PERICLES campaign"

_EGUsphere, 2025_

## Referee Comment (RC1)

**Accept, after minor revision**

The manuscript *"Profiling pollen and biomass burning particles over Payerne, Switzerland using laser-induced fluorescence lidar and in situ techniques during the 2023 PERICLES campaign"* by Gidarakou et al. presents vertical profiles of atmospheric particles obtained with a multi-channel elastic-fluorescence lidar combined with ground-based on- and offline instrumentation, adapting Veselovskii's (2021) single-channel approach for distinguishing smoke from pollen to a multi-channel system. The authors also introduce a methodology, based on Richardson (2019), that deconvolutes LIF lidar signals and compares them with reference fluorescence spectra from a pollen database to characterize pollen types. Pollen were observed near the ground up to ~2 km, exhibiting strong fluorescence backscatter coefficients $b_F$ ($2 \times 10^{-4}$ to $8 \times 10^{-4}$ $Mm^{-1}$ $sr^{-1}$ at 355 nm) and confirmed via *in situ* fluorescence measurements and Hirst trap sampling. Biomass burning particles from Canadian and German wildfires were detected higher in the atmosphere (3–5 km) with weaker $b_F$ values. Comparison across European lidar stations revealed a ~50% reduction in $b_F$ for smoke plumes transported in the free troposphere, which the authors suggest may result from photochemical aging and mixing with non-fluorescent particles. Ice-nucleating particle (INP) measurements near the ground showed a correlation between $WIBS_{ABC}$ aerosols and INPs at -14 °C, indicating bioaerosol contributions at warmer temperatures. No such correlation was observed at -20 °C, where the authors suggest coarse dust particles are linked to INPs.

The study presents a valuable dataset and analysis in my view. From a formal perspective, the quality of the manuscript is good; it is well-written, the figures and tables are mostly clear, and the arguments are presented clearly. The manuscript seems in line with the scope of ACP. In terms of content and format, it would profit from some minor improvements as listed below. Overall, I recommend to accept the manuscript for publication in ACP after these points have been addressed.

**Specific comments:**

**Structure:** The 'Discussion and Conclusions' section begins by introducing an analysis of aged smoke particles, which seems inappropriate for this section. The corresponding results and figure should be moved to the 'Results' section, in my view.

**Figures:** Clarity and accessibility of the figures could be improved. Few points here that caught my attention: (i) The rainbow color scale used in multiple graphs is quite controversial, as it may distort and mislead due to non-uniform changes, and should therefore be avoided. (ii) Fig. 2: Panel b) is rarely recognizable in the printed version, and panel c) offers little scientific relevance, as it only shows a container without technical details (e.g., inlets or instrument placement) and appears more like a group photo. I therefore suggest removing both panels, as they add no substantial information. (iii) Fig. 5 and Fig. 8: The x-axis ticks are not consistently aligned across the subplots, which is confusing and makes it unclear whether the same time stamps are shown. (iv) Fig. 6 and Fig. 9: Subplots d)–i) are slightly misaligned. Introducing uniform distances between the individual subplots would improve overall presentation. Additionally, scaling the y-axes to match those of a), b), and c) would make the graphs easier to compare.

**WIBS data:** No thresholds are specified. Without forced-trigger measurements and appropriate correction, the data is likely overestimated. If a threshold was applied, it should be specified (e.g., 3σ) to ensure comparability.

**INP-Analysis:** The manuscript strongly associates -20 °C INPs with coarse-mode dust based on correlation with $WIBS_{Coarse}$ particles. However, no direct evidence for dust is presented; $WIBS_{Coarse}$

could also include pollen, spores, or sea salt. This statement should be toned down accordingly. In addition, the observation is currently mentioned only in the abstract and results, but not in the discussion, where it should be addressed in detail. In general, the reasoning for the INP measurements and the conclusions drawn from them could and should be more thoroughly integrated into the overall argumentation, as the INP analysis otherwise appears somewhat arbitrary.

**Fig. 10**: It is not clear why the pollen fluorescence spectra from 12/06 and 15/06 differ so strongly, despite both representing 100% *Dactylis* presence.

**Technical comments:**

- Standardise *in situ* throughout the manuscript.

- Rapid-E Instrumentation is mentioned but no data are provided; consider adding data or removing the mention.

- Use non-breaking spaces for units to avoid splitting units at line breaks.

- Line 94: Add a space after "Fig. 1(d-e)."

- Line 120: Clarify if "Swisensat" refers to Swisens Poleno.

- Line 124: Correct "Induced light-induced fluorescence".

- Lines 177–178: Define LIF only once.

- Line 205ff: Define $P_F$ as the fluorescence channel.

- Line 290ff: The description in Fig. 4 is sufficient; consider removing repeated explanation from the main text.

- Line 339: Clarify "Improper calibration has been documented"; did it occur in this study or do you refer to calibration errors in the literature?

- Line 404: Replace "approximately" with "relatively" or "fairly".

- Fig. 10a): Correct the continuous line legend to the dates 13/6–16/6.

- Line 606: There might be an error in this phrase: "Our novel methodology characterizes pollen types by deconvoluting LIF lidar signals and comparing them with reference fluorescence spectra from an extensive pollen database".

---

## Referee Comment (RC2)

[revised manuscript text omitted]

*Could the uncertainty be higher by setting the LR? Different types of aerosol can have very different LRs. In addition to ageing, air masses can be mixed.*

To obtain the vertical profile of $b_{355}$ we used the Klett inversion technique (Klett, 1985), using typical lidar ratios at 355 nm of
45 sr for aged BB particles from North American wildfires (Hu et al., 2022; Mylonaki et al., 2021; Ortiz-Amezcua et al., 2017), 50 sr for pollen particles at low altitudes (Veselovskii et al., 2022a) and 60 sr for fresh BB particles from European wildfires and pollen (Burton et al., 2012; Nepomuceno Pereira et al., 2014). The uncertainties of the retrieved $b_{355}$ values, are of the order of 10-15% (Mattis et al., 2002). The molecular backscatter coefficient at 355 and 470 nm was calculated using a standard atmosphere model by the Global Land Data Assimilation System (GLDAS) (Rodell et al., 2004); thus, the ratio $\frac{T_E}{T_F}\big|_{mol}$ for the
molecular contribution can be derived. As the particle atmospheric transmission differs little between 355 and 470 nm, we used an appropriate value of the extinction-related Ångström exponent $\gamma$ equal to 1.16 (Veselovskii et al., 2020), valid for western wildfires measured for the pair 355-532 nm (Janicka et al., 2017) to account for the spectral dependence of the aerosol extinction between wavelengths 355 and 470 nm. As the differential particle transmission for a wavelength separation between 470 and 532 nm is less than 62 nm for low to medium aerosol loads, the induced error on $\frac{T_E}{T_F}\big|_{par}$ for the particulate contribution
remains less than 6% (Gast et al., 2025).

*The error becomes of secondary order because it is the ratio of two channels that is considered.*

[revised manuscript text omitted]

We see in this Figure that the grass pollen load can reach high to very high levels in May and June, with daily average values ranging between 5.35 to 342.76 particles m⁻³ in May and from 52.13 to 271.42 particles m⁻³ in June (not shown). Moreover, *Fagus* (Beech) pollen concentration is moderate during May (maximum daily value reaching 57 particles m⁻³), while in June it is found to be very low. *Betula* (Birch) pollen concentrations still exhibit moderate values during May, (maximum values reaching 14.92 particles m⁻³), while in June the concentrations decreased to low levels. Finally, *Quercus* (Oak) pollen presented high values in May (ranging from 0.67 to 106.45 particles m⁻³) and low levels during June (values ranging from 0.82 to 38.27 particles m⁻³). It is important to note that the above-mentioned concentration values represent daily averages, the instant concentration values fluctuate strongly around those averages and can reach much higher levels (Chappuis et al., 2020). In the following sections we will present two case studies of bioaerosols profiles obtained during periods of high pollination at ground level and long- and near-range transport of BB particles originating from wildfires in Canada and Germany.

A subsection titled 'Pollen climatology of the site' would be included before 4.1.

**4.1. Case 27-29 May 2023**

The first case study concerns the period from 27 May (00:00 UTC) to 30 May 2023 (00:00 UTC), when the highest pollen concentrations were measured at ground level, during the PERICLES campaign. The 10-day backward trajectory analysis for the air masses arriving over Payerne at heights over 1.5 km a.s.l. in the period 27-29 May, was obtained by the HYSPLIT model (Fig. S1a,b,c), which confirmed that these air masses originated from wild forest fires (small red dots in Figure S1a,b,c

Do you use Hysplit in ensemble mode?
Over a period of 10 days, the cumulative errors are significant on the back trajectories, and it is preferable to use multiple runs.

[revised manuscript text omitted]

---

## Author Comment (AC1)

**Replies to Referee #3**

We would like to sincerely thank the Referee for their careful reading of our manuscript and for their constructive and helpful comments. Their suggestions have helped us to significantly improve the clarity, structure, and scientific depth of the paper. In the revised version, we have addressed all comments point by point. Changes in the text are clearly marked in the manuscript with red fonts.

Below, we provide detailed responses (*red fonts*) to each reviewer's comment (*black fonts*) and indicate where the corresponding revisions have been implemented.

**General comments:**

The authors in the present study investigated the vertical profiles of pollen grains with a multi-channel elastic-fluorescence lidar and evaluated their findings in relation to the results obtained from real-time devices like Poleno and WIBS as well as to Hirst sampler. In addition, particulate matter originating from the Canadian and German megafires were detected in the atmosphere.

The manuscript is within the aims and scope of ACP. The overall study is well designed and the results clearly presented. The manuscript is well-organized and written. Some typos have been highlighted in the text. Comments regarding specific issues are included in the manuscript.

My recommendation is to be accepted for publication after the authors address all comments.

We sincerely thank the Referee for the careful reading and the constructive comments, which helped us clarify several important points regarding the interpretation of biological particle detection, the comparison of the Hirst and SwisensPoleno instruments, and the biochemical discussion of the fluorescence channels. All comments have been addressed in the revised manuscript, as detailed below.

**Specific comments:**

**Comment:**

How it is guarantied that biological particles are not excluded? Fungal spores for example are characterized by an extremely wide range of shapes and sizes.

**Reply:**

We agree and have corrected the statement. "Note that the first stage of the filter is prone to false-negative when considering fungal spores that show a greater morphological variability than pollen."

**Comment:**

The reduced time resolution (a comment I do not agree with) should be compared against the limitations of the Poleno described above and the WIBS below, limitations that are not mentioned at all. Poleno excludes a large portion of bioparticles because of size and shape (see previous relevant comment) and WIBS practically confirms the biological origin of particles sized 0.5-30µm.

**Reply:**

Reduced time resolution is a fact: spreading of the impacting particles on the Hirst sampler band corresponds to around 2 mm which amounts to around two hours uncertainty for mere physical reasons. This has to be compared with the resolution of the timing of measurements of individual particles at the millisecond level by the Swisens Poleno.

We however agree that comparison of instruments can be more balanced (see our reply to the comment below and the corresponding changes to the manuscript).

**Comment:**

Tryptophan, NAD(P)H and riboflavin are essential elements present in all organisms not just microorganisms!

**Reply:**

We agree and have corrected the statement to reflect this broader biological presence: "The three channels can detect different biologic fluorophores—tryptophan-containing proteins, NAD(P)H co-enzymes, and riboflavin (Kaye et al., 2005; Savage et al., 2017)—which are ubiquitous in living organisms, including microbes and pollen." (L162–164)

We have also corrected minor typographical errors and regarding Figure 4, in the calendar, Grasses should be mentioned, and not Dactylis glomerata. With the microscope, we count all grass pollen types as "Grasses". Similarly, it would be better to use Genus names only, as it is not possible to differentiate the pollen of the different species within these genera.

**Comment:**

It is not clear the reason for not detecting. Although Betula and Fagus have similar shapes the size is much different. How is it possible Poleno not to detect neither one?

**Reply:**

We have clarified this in the revised manuscript. "The SwisensPoleno was not able to achieve a sufficient number of precise detections for the label to be activated (Crouzy et al., 2022)." (L344-345)

In line 346, we did some corrections: Hirst sampler has undoubtedly limitations but it certainly does not perform poorly in the hourly level. Certainly, some particles may not be trapped when impacted on the adhesive but that does not mean it has a poor hourly resolution. The transverse traverses give hourly data with very good taxa representation and concentration but they are time consuming and usually avoided. On the other hand, Poleno failed to register Betulaceae and Fagaceae. The explanation given for it is inadequate. Furthermore, in previous section it is mentioned that 4 horizontal lines were measured. This way we have mean daily concentration. In Fig 5 hourly data are presented.

This paragraph was revised significantly to a more accurate comparison of the sampler's performance. We have reformulated the paragraph in the line of the suggestions by the referee.

"The Hirst trap has a lower temporal resolution compared to the SwisensPoleno: in the operational longitudinal scanning mode used at MeteoSwiss it shows limitations at the hourly level due to limited sampling efficiency and broad band spreading. However, especially for high pollen concentrations, Hirst samplers can give a hint on the sub daily evolution of the pollen concentration. Hirst performance improves at the daily level but deviations between Hirst traps operating in parallel can still be observed (Adamov et al., 2024). Notably (Oteros et al., 2017). improper calibration of Hirst flow has been documented, which may lead to deviations reaching up to 72% These factors highlight the limitations in Hirst precision and reliability, especially for fine temporal analyses. Those limitations are currently still balanced by the more precise discrimination capabilities of Hirst measurements compared to the commercially-available automatic instruments." (L346-353).

---

## Author Comment (AC2)

**Replies to Referee #1**

We would like to sincerely thank the Referee for their careful reading of our manuscript and for their constructive and helpful comments. Their suggestions have helped us to significantly improve the clarity, structure, and scientific depth of the paper. In the revised version, we have addressed all comments point by point. Changes in the text are clearly marked in the manuscript *with red fonts*.

Below, we provide detailed responses (*red fonts*) to each reviewer's comment (*black fonts*) and indicate where the corresponding revisions have been implemented.

**General comments:**

The manuscript "Profiling pollen and biomass burning particles over Payerne, Switzerland using laser-induced fluorescence lidar and in situ techniques during the 2023 PERICLES campaign" by Gidarakou et al. presents vertical profiles of atmospheric particles obtained with a multi-channel elastic-fluorescence lidar combined with ground-based on- and offline instrumentation, adapting Veselovskii's (2021) single-channel approach for distinguishing smoke from pollen to a multi-channel system. The authors also introduce a methodology, based on Richardson (2019), that deconvolutes LIF lidar signals and compares them with reference fluorescence spectra from a pollen database to characterize pollen types. Pollen were observed near the ground up to ~2 km, exhibiting strong fluorescence backscatter coefficients bF (2 × 10-4 to 8 × 10-4 Mm-1 sr-1 at 355 nm) and confirmed via *in situ* fluorescence measurements and Hirst trap sampling. Biomass burning particles from Canadian and German wildfires were detected higher in the atmosphere (3–5 km) with weaker bF values. Comparison across European lidar stations revealed a ~50% reduction in bF for smoke plumes transported in the free troposphere, which the authors suggest may result from photochemical aging and mixing with non-fluorescent particles. Ice-nucleating particle (INP) measurements near the ground showed a correlation between WIBSABC aerosols and INPs at -14 °C, indicating bioaerosol contributions at warmer temperatures. No such correlation was observed at -20 °C, where the authors suggest coarse dust particles are linked to INPs.

The study presents a valuable dataset and analysis in my view. From a formal perspective, the quality of the manuscript is good; it is well-written, the figures and tables are mostly clear, and the arguments are presented clearly. The manuscript seems in line with the scope of ACP. In terms of content and format, it would profit from some minor improvements as listed below. Overall, I recommend to accept the manuscript for publication in ACP after these points have been addressed.

We sincerely thank the Referee for the constructive review and for the overall very positive assessment of our manuscript. The Referee's feedback, especially regarding the scientific relevance, clarity of the manuscript, and the robustness of the results, is deeply appreciated. We have also addressed all specific and technical comments with detailed revisions and responses, as outlined below.

**Specific comments:**

**Structure**

**Comment:**

The "Discussion and Conclusions" section begins by introducing an analysis of aged smoke particles, which seems inappropriate for this section. The corresponding results and figure should be moved to the "Results" section.

**Reply:**

We agree with the reviewer. The corresponding analysis and figure have been moved to the *Results* section, and the *Discussion and Conclusions* section has been revised accordingly.

**Figures**

**Comment:**

Clarity and accessibility of the figures could be improved. Few points here that caught my attention: (i) The rainbow color scale used in multiple graphs is quite controversial, as it may distort and mislead due to non-uniform changes, and should therefore be avoided. (ii) Fig. 2: Panel b) is rarely recognizable in the printed version, and panel c) offers little scientific relevance, as it only shows a container without technical details (e.g., inlets or instrument placement) and appears more like a group photo. I therefore suggest removing both panels, as they add no substantial information. (iii) Fig. 5 and Fig. 8: The x-axis ticks are not consistently aligned across the subplots, which is confusing and makes it unclear whether the same time stamps are shown. (iv) Fig. 6 and Fig. 9: Subplots d)–i) are slightly misaligned. Introducing uniform distances between the individual subplots would improve overall presentation. Additionally, scaling the y-axes to match those of a), b), and c) would make the graphs easier to compare.

**Reply:**

(i) We acknowledge this point. Unfortunately, for these specific plots, alternative color maps did not preserve the necessary contrast to distinguish signal variations. Therefore, we maintained the existing color scale but improved the colorbar normalization and annotations for better readability. (ii–iv) All issues regarding alignment, scaling, and figure clarity have been corrected in the revised manuscript. The updated figures can be found in the new version.

**WIBS Data**

**Comment:**

No thresholds are specified. Without forced-trigger measurements and appropriate correction, the data is likely overestimated.

**Reply:**

We have clarified this point in the revised manuscript (L169–171):

"Here, we used the averaged forced trigger signal plus  $9\sigma$  to subtract background noise for calculating the fluorescent signal of sampled aerosol particles, instead of  $3\sigma$  used in other studies (Savage et al., 2017), to minimize the influence from interfering particles."

**INP Analysis**

**Comment:**

The manuscript associates –20 °C INPs with coarse-mode dust based on correlation with WIBSCoarse particles, but this could also include pollen, spores, or sea salt. The statement should be toned down, and the INP results should be better integrated into the discussion.

**Reply:**

We thank the reviewer for this valuable comment. We have revised the relevant text to soften the interpretation and to better integrate the INP analysis into the overall discussion. Specifically, L488–489 was corrected to:

"In addition, the INPs at -20 °C are correlated with WIBScoarse particles (Fig. S4c, d), showing  $\rho_{Pearson} = 0.72$  (pPearson = 0.01) and  $\rho_{Spearman} = 0.52$  (pSpearman = 0.10), suggesting that coarse-sized particles such as soil dust and pollen contribute to the observed INPs at -20 °C."

We also added the following in L189-191: "The freezing ability of aerosol particles determines their removal processes, lifetime, and climate impacts. Therefore, it is important to evaluate the freezing and cloud interaction abilities of the observed aerosol sources."

In addition, relevant aspects of the INP discussion have been expanded in the revised manuscript.

**Figure 10**

**Comment:**

It is not clear why the pollen fluorescence spectra from 12/06 and 15/06 differ so strongly, despite both representing 100% *Dactylis* presence.

**Reply:**

We clarified this point in the revised text (L563): "Note that differences between the in vivo and lidar-derived spectra can result from the contribution of other fluorescing particles or other grass pollen species (*Dactylis* is used as a proxy for the grass pollen family). The absence of a tail at longer wavelengths, however, suggests a dominant contribution from grass pollen."

**Technical Comments:**

1. Standardise in situ throughout the manuscript.

The term in situ has been standardized throughout the text.

2. Rapid-E Instrumentation is mentioned but no data are provided; consider adding data or removing the mention.

References to the Rapid-E instrument have been removed, as no corresponding data are presented.

3. Use non-breaking spaces for units to avoid splitting units at line breaks.

Thank you, corrected.

4. Line 94: Add a space after "Fig. 1(d-e)."

A space was added.

5. Line 120: Clarify if "Swisensat" refers to Swisens Poleno.

Minor typographical corrections have been implemented (spaces after "Fig. 1(d-e)", clarification that "Swisensat" refers to *Swisens Poleno*

6. Line 124: Correct "Induced light-induced fluorescence".

Correction of "Induced light-induced fluorescence", and single definition of LIF).

7. Lines 177–178: Define LIF only once.

**Corrected.**

8. Line 205ff: Define PF as the fluorescence channel.

PF has been defined as the fluorescence channel.

9. Line 290ff: The description in Fig. 4 is sufficient; consider removing repeated explanation from the main text.

Repetitive text following Fig. 4 has been shortened.

10.Line 339: Clarify "Improper calibration has been documented"; did it occur in this study or do you refer to calibration errors in the literature?

Regarding Line 339 ("Improper calibration has been documented"), this sentence now clarifies that we refer to calibration errors reported in the literature. We also added the sentence:

"Those limitations are currently still balanced by the more precise discrimination capabilities of Hirst measurements compared to the commercially available automatic instruments." (Line 355-356)

11. Line 404: Replace "approximately" with "relatively" or "fairly".

"Approximately" was replaced with "relatively" (Line 422).

12 Fig. 10a): Correct the continuous line legend to the dates 13/6–16/6.

The legend in Fig. 10a has been corrected to indicate the continuous line refers to 12-15 June.

13 Line 606: There might be an error in this phrase: "Our novel methodology characterizes pollen types by deconvoluting LIF lidar signals and comparing them with reference fluorescence spectra from an extensive pollen database".

The sentence at Line 606 (now 623) has been rephrased for clarity.

---

## Author Comment (AC3)

**Replies to Referee #2**

We would like to sincerely thank the Referee for their careful reading of our manuscript and for their constructive and helpful comments. Their suggestions have helped us to significantly improve the clarity, structure, and scientific depth of the paper. In the revised version, we have addressed all comments point by point. Changes in the text are clearly marked in the manuscript *with red fonts*.

Below, we provide detailed responses (*red fonts*) to each reviewer's comment (*black fonts*) and indicate where the corresponding revisions have been implemented.

**General comments:**

I found this article very interesting. It presents a novel approach that uses lidar measurements to perform coarse pollen speciation in the lower troposphere.

The methodology is clearly explained, and the results are convincing. However, there is insufficient discussion of uncertainties. One important point is calibration. Certain aspects need to be clarified for readers so that they can rely on and use this work.

In my opinion, this article is entirely appropriate for ACP and can be published once the additional information requested has been provided. My corrections and questions are in the body of the article (in supplement).

This article provides a new perspective on the use of lidar technology and requires only minor revisions.

We thank the reviewer for the careful reading and helpful suggestions that improved the overall clarity and consistency of the manuscript.

We have defined all acronyms and terms at first use, added missing parentheses where appropriate, and corrected typographical and syntactic errors. Figure 1 has been updated to correctly position the "30°" label. In addition, many sentences that have already stated, have been omitted, and spacing inconsistencies have been corrected throughout the text.

**Specific comments:**

**Comment:**

If the density is lower, that is normal. Some information is missing.

**Reply:**

That's right. Corrected to: "During this campaign, pollen particles were detected near ground (up to 2 km height), showing strong fluorescence backscatter coefficients bF at 355 nm (up to 8 x 10-4 Mm-1sr-1)." (L23-24)

**Comment:**

Could it also be related to dispersion during transport?

**Reply:**

Yes, that's right, the corrected paragraph is: ".... mixing with non-fluorescent particles and air mass dispersion processes." (L33)

**Comment:**

How many? (regarding the Temperature)

**Reply:**

Corrected to: "The latter can occur at relatively warm sub-zero temperatures ( $\sim$  -2 to -10 °C), and thus bioaerosols can initiate ice multiplication that leads to rapid glaciation, storm intensification and extreme precipitation (Gao et al., 2025; Lohmann et al., 2016; O'Sullivan et al., 2015)."

**Comment:**

Dust and BB aerosols are not particularly organic.

**Reply:**

Agreed, the sentence was modified:" The main objective of this campaign was to understand the spatio-temporal variability of different types of bioaerosols containing pollen, BB and dust within the Planetary Boundary Layer (PBL) and lower free troposphere aloft (typically up to 2-5 km a.s.l.)"

**Comment:**

Could the uncertainty be higher by setting the LR? Different types of aerosol can have very different LRs. In addition to ageing, air masses can be mixed.

**Reply:**

Correct. Different aerosol types have different LRs, therefore, we performed a sensitivity analysis for the three aerosol types. Added to L232–233:

"The uncertainties of the retrieved b355 values, are of the order of 25-30%, based on a sensitivity analysis performed for these three different types of aerosols."

**Comment:**

The error becomes of secondary order because it is the ratio of two channels that is considered.

**Reply:**

Correct, thanks for noticing, added to L240: ".... the induced second order error on  $\frac{T_E}{T_F}|_{par}$  for the particulate contribution remains less than 6% (Gast et al., 2025)."

**Comment:**

I understood that an elastic channel had been used. Could you confirm this?

**Reply:**

Yes, we confirm. The paragraph was modified (L247–251): "Furthermore, to equalize (calibrate) the PMT sensitivities, we installed the PMT from the fluorescence channel to the so called "Raman" one (Veselovskii et al., 2020); then, by adjusting the voltage supply, we obtained the same signal intensity  $P_{F355}$  as the elastic one  $P_{E355}$  at the analog channel at 355 nm. This equalization (calibration) ratio can be expressed by the ratio  $\left[\frac{P_{E355}}{P_{F355}}\right]_{an,cal}$  both at 355 nm, along the whole range of the analog channel and is found to be  $\left[\frac{P_{E355}}{P_{F355}}\right]_{an,cal} = 0.12$ ."

**Comment:**

What level of uncertainty is associated with such a calibration? Can the calibration of the elastic channel be verified over an altitude range where molecular scattering dominates?

**Reply:**

We only measured in altitudes where bioaerosols dominated. Revised to (L256-257) hus we transformed the sentence into:

"Thus, in our LIF lidar system we acquired a value of F=65 when gluing the  $P_{Fpc}$  (in MHz) to  $P_{Fan}$  (in mV) within the altitude range where the bioaerosols dominate."

**Comment:**

What is the deconvolution function? It's not clear.

**Reply:**

Apologies for the confusion, its actually the method of spectral decomposition. Thus, we transformed the L266-267: "The method is based on the spectral decomposition of the LIF lidar signals obtained by a multichannel lidar detector to determine the contribution from each taxon."

**Comment:**

I am wondering whether this is really a deconvolution in the mathematical sense of the term, or whether it is proximity recognition using a cost function?

**Reply:**

Please see the above response. Please see the clarification above — it is a spectral decomposition method, not a mathematical deconvolution.

**Comment:**

The introduction does not address health issues; perhaps a few lines should be added.

**Reply:**

Thank you for noticing, we added a few lines in Introduction regarding health issues: "This prolonged exposure has been linked to a rise in allergic respiratory diseases such as allergic rhinitis and asthma, affecting millions of people worldwide and imposing a growing public health burden, especially in urban environments where interactions with air pollutants can further intensify symptoms (Buters et al., 2018; D'Amato et al., 2020). Understanding the

dynamics of pollen emission, transport, and transformation in the atmosphere is thus crucial for improving allergy forecasts and assessing health risks under changing climatic conditions." (L53-57).

**Comment:**

This paragraph should be separated between the introduction and Section 2.

**Comment:**

A subsection titled 'Pollen climatology of the site' would be included before 4.1.

**Reply:**

Implemented as suggested. A new subsection titled *Pollen Climatology of the Site* has been added before Section 4.1.

**Comment:**

Do you use Hysplit in ensemble mode? Over a period of 10 days, the cumulative errors are significant on the back trajectories, and it is preferable to use multiple runs.

**Reply:**

Thank you for the suggestion, we added it in the Supplement.

**Comment:**

Temperature is one factor to consider, but shouldn't we also pay attention to the importance of solar radiation?

**Reply:**

Correct. Added to L328-329: "It is well documented that grass pollen concentration levels generally peak during daytime (generally higher solar irradiance and temperature levels compared to nighttime) and often tend to increase after rainfall (Kelly et al., 2013; Sabo et al., 2015)."

**Comment:**

These two meteorological variables are generally anticorrelated.

**Reply:**

We agree. Many field studies (including CALISHTO at Mt Helmos) observed anticorrelations between T and RH, although situations with high T and low RH can still occur.

**Comment:**

What are the error bars on these measurements?

**Reply:**

The metrological quantification of both manual and automatic measurement is an ongoing effort (https://www.bioairmet.ptb.de/). We are not yet able to provide precise quantification of the errors.

**Comment:**

Pollen is generally not included in PM2.5 or even PM10 measurements. How can this data be linked to pollen concentrations?

**Reply:**

Particles smaller than 10 micrometers are filtered out after the trigger laser stage in the Swisens Poleno. Conversely, Pollen grains are too large to be included in PM10 measurements. Those measurements do not overlap.

**Comment:**

Could you elaborate on the significance of ion chromatography in this study?

**Reply:**

Ion chromatography is used to identify and quantify major inorganic ions, providing chemical context that helps distinguish aerosol sources (e.g., biomass burning, pollution, or secondary formation) relevant to interpreting lidar observations.

**Comment:**

For the first case study, there is no INP. It may be better to introduce the differences between the two case studies.

**Reply:**

INP measurements were not available for 27–30 May. The two days with INP data are 24 and 31 May, and this distinction has been clarified in the text.

**Comment:**

Where would they come from?

**Reply:**

Coarse-sized particles could be soil dust and pollen, or larger sized fungal spores originating from local forest and agriculture lands. We do not have Dust event on these days. Added in L484-485: "... which suggests coarse-sized particles, such as soil dust, pollen and large-sized fungal spores contribute to the observed INPs at  $-20^{\circ}$ C."

**Comment:**

Is the consideration of RH related to the hygroscopicity of the particles and the associated decrease in fluorescence?

**Reply:**

Correct. Added to L582-585: "Along this transport the  $b_F$  values passed from ~4.5 x  $10^{-4}$  Mm-1sr-1 to 2.0 x  $10^{-4}$  Mm-1sr-1, now showing a ~50% reduction on the  $b_F$  values, indicating again a possible mixing with non-fluorescent (e.g. continental polluted) aerosols of different hygroscopicity values at lower heights."

**Comment:**

It should be emphasized in the abstract that this is a new approach which reliably provides information on pollen speciation in the lower troposphere.

**Comment:**

Always in the abstract, it is fair to say that this is an original approach.

**Reply:**

Thank you for noticing. Added to L21-L23: "This original approach provides, for the first time in this region, reliable information on pollen speciation aloft, bridging the gap between ground-based sampling and remote sensing observations."